# Immune correlates of anti-BCMA CAR-T products idecabtagene vicleucel and ciltacabtagene autoleucel in a real-world cohort of patients with multiple myeloma

We performed the first in-depth, comparative and prospective biomonitoring of Multiple Myeloma (MM) patients ($N = 39$) receiving ciltacabtagene auto-leucel (cilta-cel) or idecabtagene vicleucel (ide-cel) chimeric antigen receptor T cells (CAR T) in the real-world setting. In cilta-cel patients response rates were higher and atypical neurotoxicities/infections more frequent. Peak CAR T counts were significantly higher in cilta-cel patients, driven by $CD4^+$ CAR expansion, correlating with clinical responses. Expansion of cilta-cel cells was associated with higher CAR and CD27 expression while, in contrast to ide-cel, there was no correlation between TIM3 expression and CAR T proliferation. Cilta-cel CAR T expansion was followed by a CAR-specific switch from proliferation-associated genes to genes/surface markers indicating effector/memory function. The longer persistence of cilta-cel CAR T was associated with increased IL-7R expression; in vitro data showed persistent antigen-independent activation and higher metabolic activity of cilta-cel vs. ide-cel CAR T. Among cilta-cel-treated patients experiencing atypical neurotoxicities, central nervous system (CNS)-infiltrating, effector-type CAR T presented a distinct inflammatory phenotypic/cytokine-expression profile. This in-depth biomonitoring report following real-world cilta-cel or ide-cel highlights intrinsic biological differences between BCMA-targeting CAR T products, potentially explaining differences in clinical activity and toxicity. Our findings may guide optimization of cellular immunotherapy strategies in MM.

Chimeric antigen receptor (CAR) T-cell (CAR T) therapies have revolutionized the care of patients with multiple myeloma (MM)[1]. Recently, the first two B cell maturation antigen (BCMA)-targeted CAR T for MM were approved, idecabtagene vicleucel (ide-cel)[2] and ciltacabtagene autoleucel (cilta-cel)[3]. Unfortunately, most MM patients receiving this first generation of approved BCMA-targeted CAR T will experience toxicities and eventually relapse, highlighting an urgent need to optimize CAR T approaches.

While clinical trials typically include correlative biomonitoring, mechanistic associations with clinical response, toxicity and tumor escape are not routinely assessed in the real-world setting. This makes it difficult, if not impossible, to further optimize CAR T cell treatments for the average MM patient. In addition, the identification of meaningful biomarkers may support the selection of the most promising treatments for individual patients, the determination of the most appropriate dose levels, the design of ideal treatment sequencing, and timing of retreatment with antigen-specific therapy even before occurrence of relapse/progression. Therefore, we set out to perform the an in-depth and prospective analysis of immune responses to treatment with either cilta-cel or ide-cel in the

✉e-mail: datanackovic@som.umaryland.edu

real-world setting and correlate those with clinical responses and toxicities.

## Results

### Patients receiving cilta-cel experience a toxicity profile distinct from those receiving ide-cel in the real-world setting

The clinical characteristics of MM patients (*N* = 39) enrolled in our prospective biomonitoring study 2043GCCC are shown in Table 1. Of 39 patients, 23 had received cilta-cel and 16 ide-cel; all outside of clinical studies. Comparing both patient groups, there were no differences in gender, age, and race. Neither were there differences in disease characteristics, prior lines of therapy, or beta-2 microglubulin serum levels as an expression of tumor burden (median 2.7 mg/L [95% CI 2.3–3.9] vs. 2.8 [95%CI 2.1–4.3]. The only significant difference was a higher proportion of patients in the cilta-cel group (Table 1) receiving bridging therapy (78.3% vs. 43.8%). However clinical responses to bridging therapy accurred in only 20% of cilta-cel patients and in the combined patient groups there was no significant difference in terms of response rates to CAR T treatment between patients who had received bridging therapy versus those who had not. There were also no statistically significant group-differences in overall response rates (ORR) or rates of complete responses (CR) (Table 1). Analyzing CAR T-related toxicities, we found similar rates of cytokine release syndrome (CRS) and classical immune effector cell-associated neurotoxicity syndrome (ICANS). In contrast, we detected a higher rate of post-treatment infection (78.3% vs. 12.5%) in the cilta-cel group (Table 1). Most infections were bacterial (Supplemental Table 4) and were typically caused by gram-negative bacteria or Clostridium difficile. In terms of the nature of the bacterial infections our cilta-cel patients had one or more of the following: bacteremia (6/14; 43%), colitis (5/14; 36%), urinary tract infection (5/14; 36%), pneumonia (2/14; 14%), and deep tissue infection (1/14; 7%). Importantly, the cilta-cel group also showed a higher number of non-ICANS neurotoxicities (26.1% vs. 0.0%) and. Most atypical neurotoxicities consisted of facial nerve paralyses followed by Parkinson-like syndromes (Supplemental Table 3).

### Patients receiving cilta-cel show a more pronounced peak expansion and CAR T persistence compared to patients receiving ide-cel

Comparing in vivo CAR-T expansion between patient groups, we found that total CAR-T peak expansion was significantly higher for cilta-cel vs. ide-cel (Fig. 1A+F). At the timepoint the starting material for CAR T manufacturing was collected there was no difference in the CD4 + / CD8+ composition of the patients' T cells (Fig. 1E). However, after cell transfer of the final product into the patient, CD8$^+$ CAR-T expansion was similar for both groups (Fig. 1C+F) but CD4$^+$ CAR-T expansion was much more pronounced in the cilta-cel group (Fig. 1B+F). Similarly, cilta-cel recipients showed more pronounced expansion of CD4$^+$CD8$^+$ double-positive CAR T (Fig. 1D+F). Overall, peak levels for each of the three CAR-T subtypes occurred significantly later in the cilta-cel group (Fig. 1F). Consequently, at Day +7, total numbers of CAR T were higher in ide-cel than cilta-cel recipients (Fig. 1G) while at all subsequent timepoints up to 3 months after infusion, CAR T levels were higher in the cilta-cel group (Fig. 1G). ANOVA revealed a strong group effect over time on peak total CAR-T (p < 0.0001) and CD4$^+$CAR-T (p < 0.0001) cell counts, but not on peak CD8$^+$ or CD4$^+$CD8$^+$ CAR-T counts (Fig. 1A–D). As a result of higher peaks and longer persistence, the area-under-the-growth curve was greater among cilta-cel vs. ide-cel recipients (p < 0.05) (Fig. 1A). This change in T cell proportions was reflected by more pronounced increases in absolute white blood cell, lymphocyte and total T cell counts (Fig. 2A) and in absolute CAR-T cell counts (Fig. 2B) in the cilta-cel group at the proportional CAR T peak timepoint. Consequently, absolute T cell and CAR T peak numbers were significantly higher in patients receiving cilta-cel vs. ide-cel (Fig. 2C).

**Table 1 | Patient Characteristics**

| Clinical characteristic | Cilta-cel *N* (%) | | Ide-cel *N* (%) | | *P* value |
|---|---|---|---|---|---|
| Total (*N* = 39) | **23** | **(59.0)** | **16** | **(41.0)** | |
| Gender | | | | | *0.440* |
| Male | 13 | (54.2) | 10 | (66.7) | |
| Female | 11 | (45.8) | 5 | (33.3) | |
| Age | | | | | *0.217* |
| ≥ 70 years | 7 | (33.3) | 8 | (50.0) | |
| <70 years | 16 | (66.7) | 8 | (50.0) | |
| Race | | | | | *0.448* |
| Black | 6 | (26.1) | 6 | (37.5) | |
| White | 17 | (73.9) | 10 | (62.5) | |
| Disease characteristics | | | | | |
| IgG / IgA / IgD / IgM / LC | 15 / 5 / 1 / 1 / 1 | | 9 / 4 / 0 / 0 / 3 | | *0.616* |
| Oligo-/Non-secretory | 4 | (17.4) | 1 | (6.3) | *0.306* |
| Extramedullary Disease | 5 | (22.7) | 3 | (18.8) | *0.767* |
| Creatinine Clearance <60 mL/min | 6 | (26.1) | 4 | (25.0) | *0.939* |
| R-ISS 2/3 | 11 | (47.8) | 6 | (37.5) | *0.522* |
| Del17p | 6 | (27.3) | 3 | (25.0) | *0.886* |
| t(4;14) | 2 | (9.5) | 1 | (7.7) | *0.855* |
| Prior lines of treatments (median/range) | 5.0 | (5.0) | 5.5 | (8.0) | *0.640* |
| Prior therapies | | | | | |
| ASCT | 15 | (65.2) | 11 | (68.8) | *0.818* |
| Lenalidomide | 22 | (95.7) | 16 | (100.0) | *0.398* |
| Pomalidomide | 21 | (91.3) | 16 | (100.0) | *0.226* |
| Bortezomib | 21 | (91.3) | 15 | (93.8) | *0.778* |
| Carfilzomib | 20 | (87.0) | 14 | (87.5) | *0.960* |
| Anti-CD38 mAb | 23 | (100.0) | 16 | (100.0) | *0.999* |
| Bridging therapy | 18 | (78.3) | 7 | (43.8) | ***0.043*** |
| Cytokine Release Syndrome | | | | | |
| Grade 1–2 | 20 | (87.0) | 14 | (87.5) | *0.960* |
| Tocilizumab use | 19 | (82.6) | 13 | (81.3) | *0.913* |
| ICANS | | | | | |
| Grade 1-2 | 5 | (22.7) | 2 | (14.3) | *0.533* |
| Grade 3 | 3 | (13.6) | 2 | (14.3) | *0.956* |
| Steroid use | 16 | (69.6) | 8 | (50.0) | *0.217* |
| Other types of neurotoxicities | 6 | (26.1) | 0 | (0.0) | ***0.064*** |
| Other complications | | | | | |
| Infection | 18 | (78.3) | 2 | (12.5) | ***<0.0001*** |
| ICU Care | 5 | (21.7) | 2 | (12.5) | *0.460* |
| Outcome | | | | | |
| Overall response rate | 21 | (91.3) | 13 | (81.3) | *0.356* |
| Complete response rate | 14 | (63.6) | 9 | (56.3) | *0.646* |
| Progression-free survival (PFS) | 18.0 months | | 16.6 months | | *0.219* |

TTo compare individual clinical characteristics between groups a Chi Square test was used. A Kolmogorov-Smirnov test was used to compare progression-free survival between groups. A *p* = 0.05 was considered statistically significant. No adjustments were made for multiple comparisons. Italic numbers indicate *p*-values for individual statistical tests and bold numbers indicate differences that were statistically significant or trended towards statistical significance.

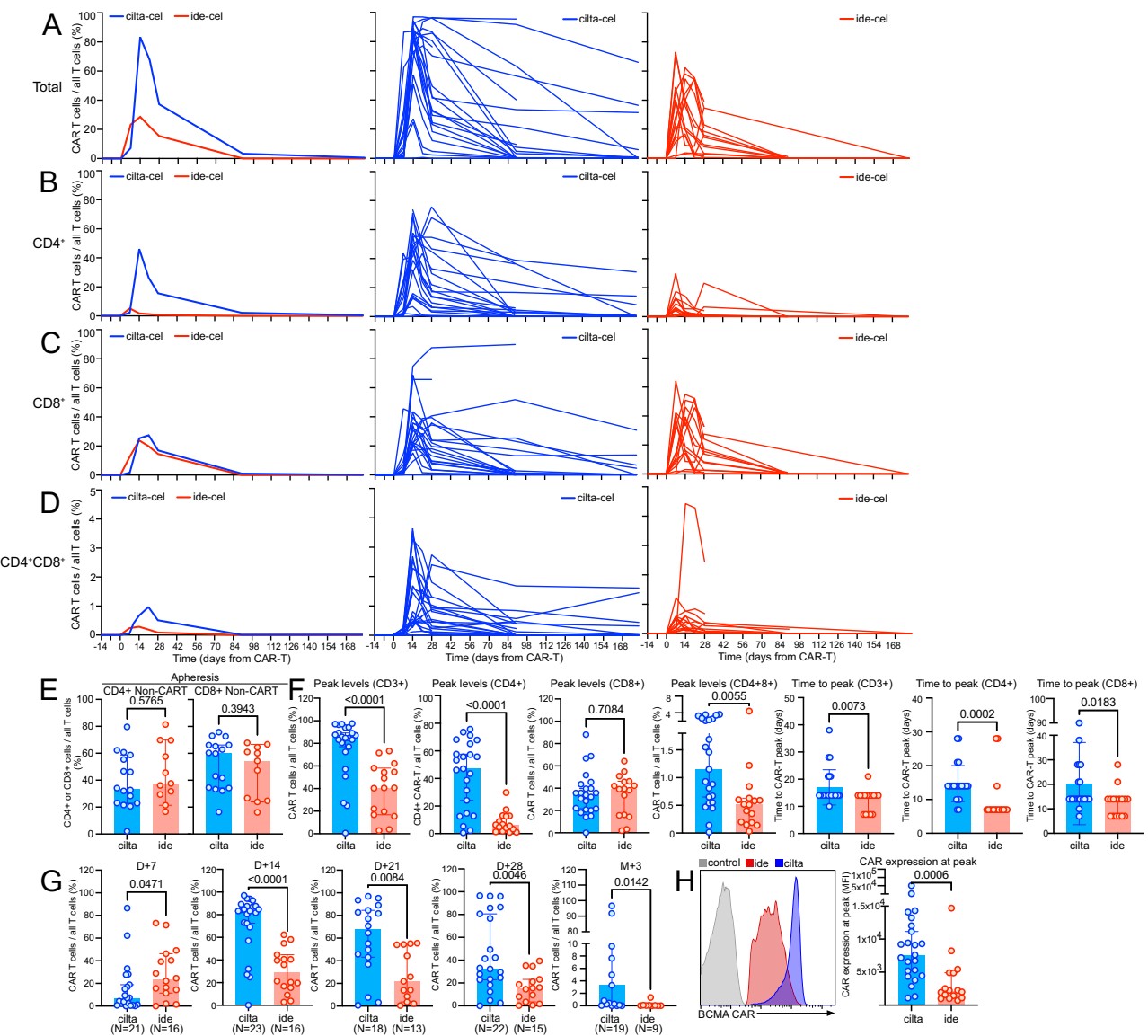

**Fig. 1 | Distinct expansion pattern of of cilta-cel vs. ide-cel CAR-T in myeloma patients in the real-world setting.** Time course of CAR T cell numbers expressed as T cells identified by surface expression of the CAR using CAR detection reagent and flow cytometry for (**A**) total CAR T, (**B**) CD4+ CAR T, (**C**) CD8+ CAR T, and (**D**) CD4+CD8+ CAR T and proportions in myeloma patients after lymphodepleting chemotherapy and CAR T cell infusion. Data are expressed as the respective CAR-T subtype per total number of T cells. Blue curves indicate cilta-cell numbers and red curves indicate ide-cell numbers. **E** Levels of CD4+ and CD8+ Non-CAR T cells in the patients' blood at the time of apheresis for cilta-cel (blue; N = 15) vs. ide-cel (red; N = 11) patients. **F** Peak CD3+ CAR T levels, peak CD4+ CAR T levels, peak CD8+ CAR T levels, peak CD4+CD8+ CAR T levels, time to peak for CD3+ CAR T, time to peak for

CD4+, and time to peak for CD8+CAR T for cilta-cel (blue; N = 23) vs. ide-cel (red; N = 16). **G** CAR T cell numbers at different timepoints post cell infusion for cilta-cel (blue) vs. ide-cel (red). **H** Surface expression levels of the respective BCMA CAR as measured by flow cytometry after staining with BCMA CAR detection reagent. The histogram shows expression levels on CD3+ CAR T in two exemplary patients (gray=unstained control, red=patient who received ide-cel, blue=patient who received cilta-cel). The bar graph on the right shows surface expression levels of the given CAR at peak in the two groups of patients (blue, N = 23; red=ide-cel, N = 16). Bar graphs indicate median values with 95% confidence intervals (CI). Statistical differences between groups were calculated using a two-sided Mann-Whitney U test. Source data are provided as a Source Data file.

We next analyzed whether CAR T cell expression of CD27 (Supplemental Fig. 1A) was associated with their proliferative activity as CD27 expression has been associated with enhanced CAR T activation and persistence[4]. We found that early after CAR T infusion, cilta-cel recipients showed much higher expression of CD27 on CD4+ and CD8+ CAR T than ide-cel recipients (Supplemental Fig. 1B); this difference was maintained through 4 weeks after treatment (Supplemental Fig. 1C+D). Combining data from both products, there was a significant positive correlation between CD8+ CAR-T peak expansion and CD27 expression at the same timepoint (Supplemental Fig. 1E).

The proliferative activity of the cilta-cel cells was often so impressive that we were alerted by our hematopathology team about the appearance of high numbers of "atypical" lymphocytes in the patient's peripheral blood (Supplemental Fig. 2). This phenomenon occurred exclusively in cilta-cel patients around the day of peak expansion ( ~ day+11). The atypical lymphocytosis (up to 95% of leukocytes), which represented CAR T as confirmed by flow cytometry, was typically comprised of large forms with ovoid to irregular nuclear contours, mature chromatin, and abundant palely basophilic and agranular cytoplasm (Supplemental Fig. 2).

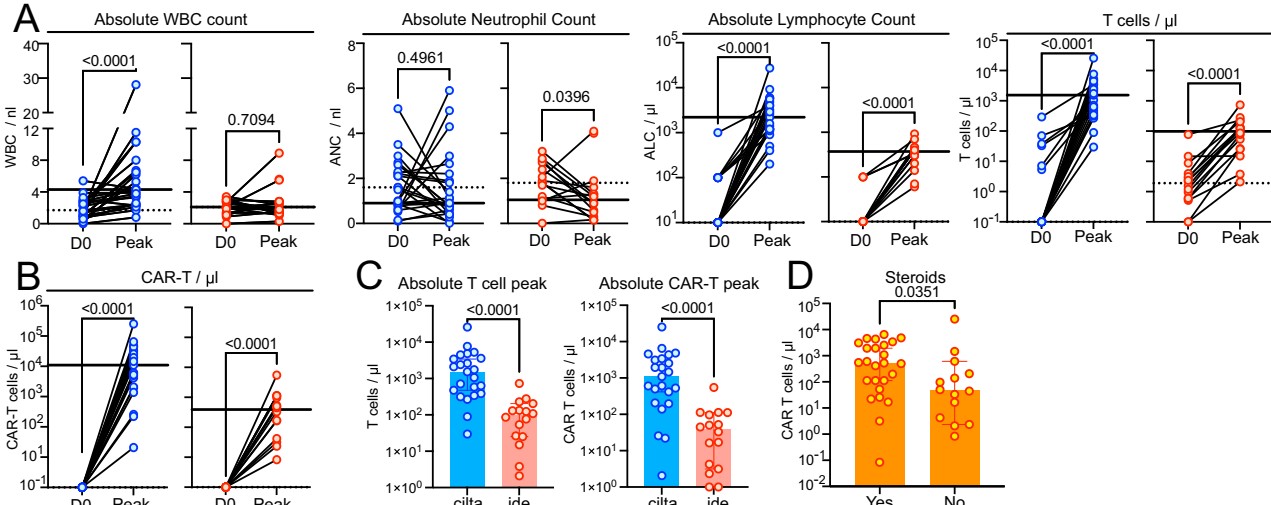

**Fig. 2 | Different patterns of immune cell reconstitution in myeloma patients post cilta-cel vs. ide-cel CAR-T. A** Dots indicate absolute counts of white blood cells (WBC), neutrophils, lymphocytes and conventional T cells in patients post cilta-cel (blue; $N = 23$) or ide-cel (red; $N = 16$) at the time of cell infusion (D0) and the time of the patient's individual CAR T peak. The dotted line and the solid lines indicate median baseline and peak levels, respectively. **B** Absolute CAR T cell counts in patients post cilta-cel (blue; $N = 23$) or ide-cel (red; $N = 16$) at the time of cell infusion (D0) and the time of the patient's individual CAR T peak. **C** Bar graphs on the left show absolute peak levels of conventional T cells and bar graphs on the right show absolute peak levels of CAR T cells in patients post cilta-cel (blue; $N = 23$) or ide-cel (red; $N = 16$), respectively. **D** Orange bar graphs show absolute counts of CAR T across cilta-cel ($N = 23$) or ide-cel ($N = 16$) patients who did or did not receive steroids post CAR T infusion. All bar graphs indicate median values with 95% confidence intervals (CI). Statistical differences between groups were calculated using a two-sided Mann-Whitney U test. Source data are provided as a Source Data file.

## Clinical responses to CAR T correlate with peak expansion rate but not CAR T dose

The overall clinical course post-CAR-T of all individual patients is shown in Fig. 3C. ORRs were higher at one month (82.6% vs. 64.3%) and 3 months (90.9% vs. 76.9%) in the cilta-cel compared to the ide-cel group although the difference was not statistically significant (Fig. 3B). There was only a non-significant difference in progression-free survival (Fig. 3D, Table 1) in cilta-cel compared to ide-cel patients (median 18.0 months vs. 16.6 months). In both groups, the depth of responses increased over the first 3 months (Fig. 3B). As per FDA labeling, patients treated with cilta-cel received significantly lower CAR T doses compared to ide-cel (Fig. 3A) but we did not observe an effect of CAR-T dose, or any other clinical characteristic, on clinical response (Table 1). Neither were CAR-T peak counts associated with any patient characteristics, biochemical markers such as peak serum ferritin level or peak C-reactive protein (CRP), CAR T dose, nor of any concomitant medications including systemic steroids (Fig. 2D). When we looked for associations between peak CAR counts with the depth of the responses separately for each patient group, we did not find any significant correlations, which was most likely due to relatively homogenous types of responses within each group and small group sizes. However, across both patient groups, there was a significant positive correlation of clinical response with peak counts of total CAR T, CD4+ CAR T and CD8+ CAR T (Fig. 3E).

## Cilta-cel and ide-cel CAR T products show distinct phenotypic and gene signatures in vivo

Differential BCMA CAR T expansion may be caused by the presence of distinct T-cell subsets and gene expression profiles due to differences in CAR constructs and/or manufacturing processes. We therefore examined T-cell phenotypes in both CAR+ and CAR- circulating T cells at each patient's peak CAR-T expansion. We found a higher proportion of effector-memory (EM) and fewer T memory stem cells (TSCM) in CAR T isolated from recipients of cilta-cel than ide-cel (Fig. 4A–D). Over time, cilta-cel CAR T cell phenotype showed a shift from EM toward terminally-differentiated EM (Supplemental Fig. 5). To substantiate the phenotypic change, we characterized cilta-cel CAR T over

time using single-cell RNA sequencing on one of the first cilta-cel patients enrolled in our study showing the typical substantial increase in effector-type CAR T. We found an analogous change in the gene signature over the course of the first 4 weeks after CAR-T infusion (Supplemental Fig. 3A). At the time of CAR T peak, ~50% of the CAR T cells, as identified by the CAR insert, expressed genes related to cellular proliferation, e.g., *Ki-67*, *DNA Topoisomerase II*, *Cyclin*, and were thus assigned to the "proliferating pool" of cells. Over the following 2-3 weeks we observed a shift within the CAR T cell population toward effector/memory T cell gene expression (Supplemental Fig. 3C-E). Interestingly and in contrast, analysis of CAR- CD4+ and CD8+ T cell subsets from the same patient/timepoint found no differences between recipients of the two CAR T products (Fig. 4E), indicating that the differential memory development in the CAR T was indeed driven by the given product's metabolic/functional characteristics and/or specific CAR signaling activity.

This was confirmed by bulk RNA sequencing comparing CAR-positive with CAR-negative CD4+ T cells (Supplemental Fig. 4) from a cilta-cel patient with strong CAR T cell expansion. Cilta-cel CAR T cells showed increased expression of genes related to effector function, such as *GNLY*, *GZMK*, *GZMA*, and *EOMES*, and a reduced expression of genes related to regulatory function, such as Helios (*IKZF2*), *FOXP3*, and CD25 (*IL2RA*), when compared to their own non-CAR T (Supplemental Fig. 4B+C). These findings further support the view that the two products have distinct functional properties leading to the development of construct-specific phenotypic signatures and function.

## Cilta-cel CAR T show a distinct constitutive activation pattern with enhanced metabolic capacity and antigen-independent cytokine secretion in vitro

To determine whether the different clinical, phenotypic, and in vivo expansion profiles of cilta-cel and ide-cel were dependent on their respective CAR constructs vs their manufacturing processes, we repeated analyses using lab-generated CAR T employing the cilta-cel[5] and ide-cel[6] CAR constructs from three healthy donors in vitro using a uniform manufacturing process (Supplemental Fig. 6B). Notably, with the uniform 10-day manufacturing process, cilta-cel CAR T manifested

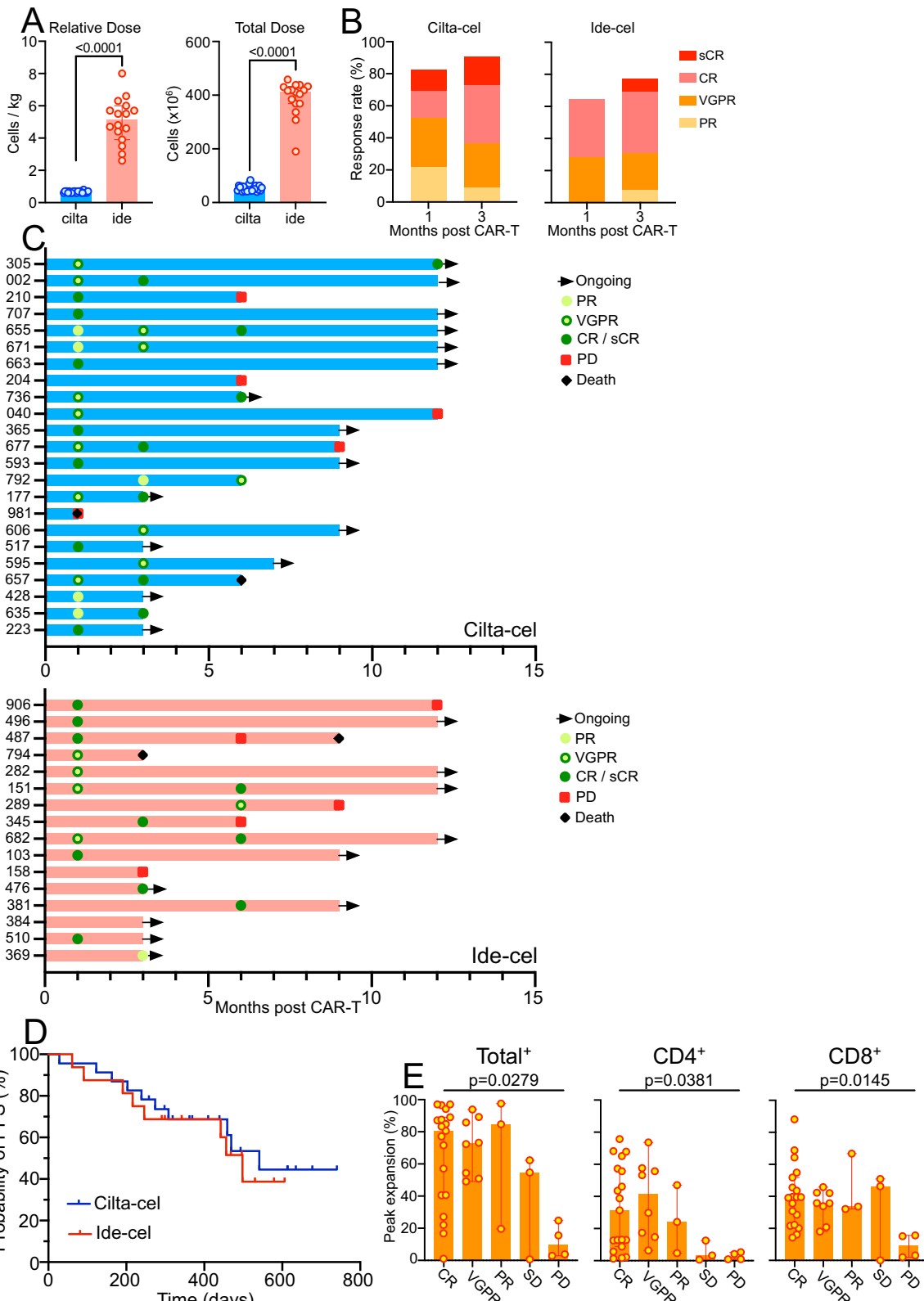

**Fig. 3 | Treatment with cilta-cel or ide-cel and correlation with clinical characteristics and outcome.** **A** Bar graphs show absolute and relative dose levels received (blue = cilta-cel, $N = 23$; red = ide-cel, $N = 16$). Statistical differences between groups were calculated using a two-sided Mann-Whitney U test. **B** Depth of response at one and three months post-CAR-T, respectively. **C** Clinical course of patients receiving cilta-cel (blue) or ide-cel (red). **D** Progression-free survival (PFS) after treatment with cilta-cel or ide-cel. **E** Depth of response at 3 months post CAR T across both patient groups ($N = 37$) in relation to peak expansion levels of total, CD4 + , and CD8 + CAR-T. Bar graphs indicate median values with 95% confidence intervals (CI). Statistical differences between response categories were calculated using a Kruskal-Wallis test (one-way ANOVA on ranks). Source data are provided as a Source Data file.

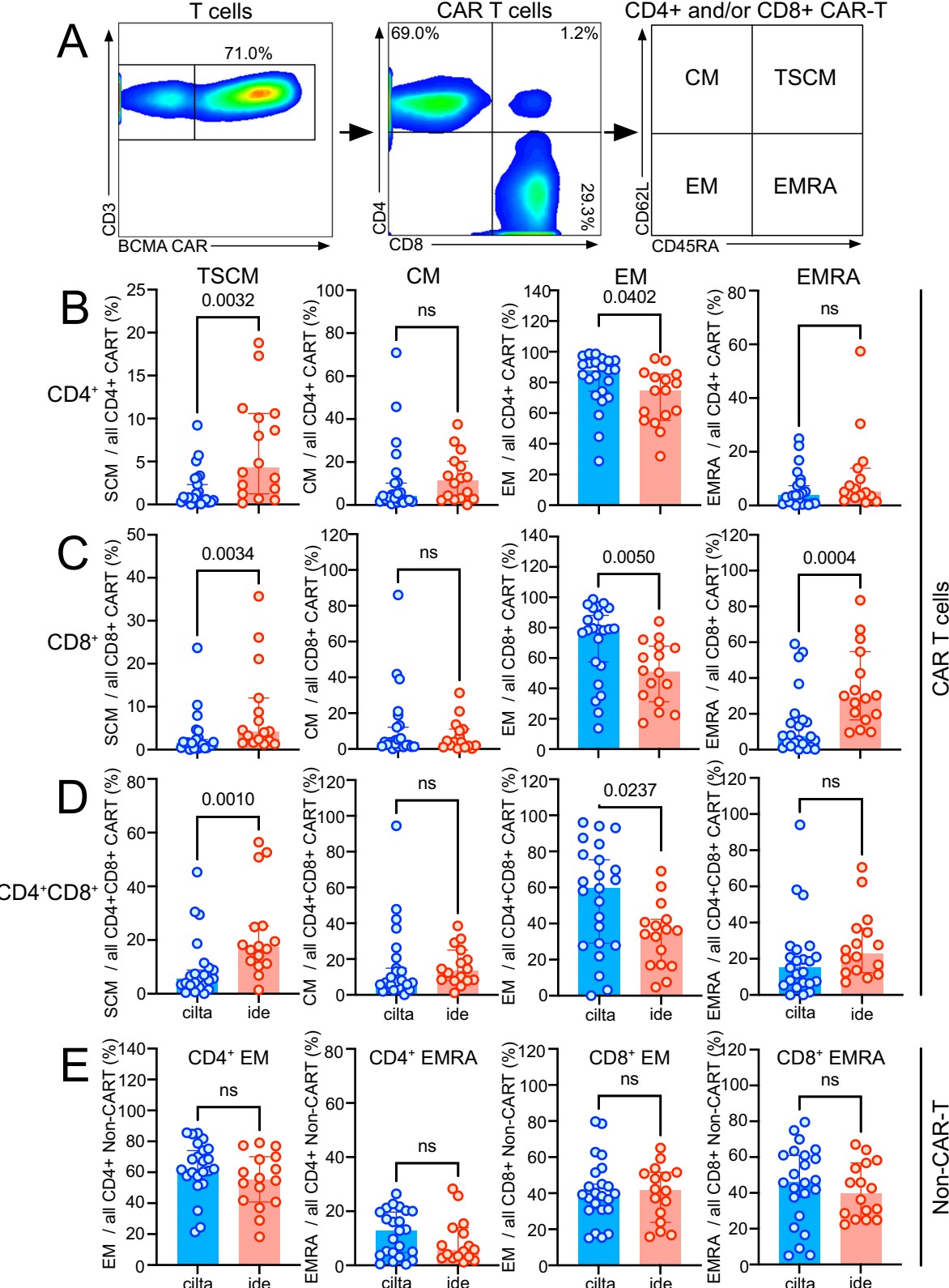

**Fig. 4 | Rapid expansion of cilta-cel CAR T is associated with a CAR-specific shift towards effector-memory T cell subtypes. A** We investigated the distribution of certain memory subtypes (CM = of central memory; TSCM = T memory stem cells; EM = effector-memory; EMRA = terminal effector memory) the different CAR T products as defined by the expression of certain memory markers as indicated. All 4 T cell memory subtypes were investigated in both patient groups (blue = cilta-cel, $N = 23$; red = ide-cel, $N = 16$) at the time of the individual CAR T peak. Results are shown for (**B**) CD4[+], (**C**) CD8[+], (**D**) CD4[+]CD8[+] CAR T cells and for (**E**) CD4[+] non-CAR T from the same patient/timepoint. Bar graphs show percentages of the given memory subtype out of all CD4[+], CD8[+], CD4[+]CD8[+] CAR T cells or CD4[+] non-CAR T in patients post cilta-cel (blue, $N = 23$) or ide-cel (red, $N = 16$), respectively. Bar graphs indicate median values with 95% confidence intervals (CI). Statistical differences between groups were calculated using a two-sided Mann-Whitney U test. Source data are provided as a Source Data file.

lower expansion and viability compared to ide-cel (Supplemental Fig. 6A), in concordance with greater expression of pro-apoptotic TRAIL/Annexin V (Supplemental Fig. 6F+G). With the cilta-cel CAR, the end-product evidenced significantly higher CAR surface density as determined by mean fluorescence intensity (Supplemental Fig. 6C) and a higher CD4/CD8 ratio (Supplemental Fig. 6E), despite similar transduction efficiency (Supplemental Fig. 6D). In addition, during overnight culture in cytokine-free media and absence of BCMA antigen, cilta-cel CAR products spontaneously secreted significantly more IL-2 and TNF-α than ide-cel, whereas both cilta-cel and ide-cel CAR products spontaneously produced IFN-γ (Supplemental Fig. 6I). At the time this enhanced cytokine production was observed, the cilta-cel CAR T expressed an effector-memory phenotype, the same memory type that was preferentially observed for cilta-cel during peak expansion (Supplemental Fig. 6H). Finally, cilta-cel CAR products demonstrated significantly greater mitochondrial metabolic capacity as measured by oxygen consumption rate (Supplemental Fig. 6J). These data indicate that the differential expansion patterns, phenotypic signatures, and perhaps even clinical characteristics may result from differences in the respective CAR constructs.

### Long-term persistence of cilta-cel is associated with a distinct effector-memory phenotype with upregulation of negative regulatory markers and IL-7Rα

As shown above (Fig. 1A–D), we exclusively observed long-term persistence of CAR T in the cilta-cel recipients; we next explored whether early CAR T exhaustion could be involved in limiting persistence. Remarkably, we found the highest levels of negative regulatory markers in cilta-cel and ide-cel-treated patients across T cell subsets immediately after infusion. This initial upregulation was followed by a decrease in these surface molecules over the following weeks (Fig. 5A, B). In the ide-cel group, we found that an "early" peak in TIM3 alone (Fig. 5D) or in combination with PD-1 and LAG-3 (Fig. 5E) predicted reduced CAR T-cell expansion. We did not observe a correlation between expression of the negative regulatory markers and CAR-T expansion in the cilta-cel group (Fig. 5D and Supplemental Figure 7).

In addition to a potentially reduced exhaustion, response to pro-proliferative soluble factors may contribute to long-term CAR-T persistence. Expression of CD127 (IL-7Rα) has been shown to be crucial for survival of long-lived memory T cells and protection from exhaustion[7–9]. We found that the long-term persistent cilta-cel CAR T had markedly upregulated expression of CD127 compared to CAR T isolated at earlier timepoints (Supplemental Fig. 8A), with a statistically significant upregulation specifically in the CD4+ CAR-T subset (Supplemental Fig. 8B).

### CNS infiltration by highly activated effector memory-type CAR T correlates with atypical neurotoxicity

Based on our observation of constitutional activation of cilta-cel CAR T associated with pronounced expansion and prolonged persistence of effector-type cells, we next asked whether the increased rate of neurotoxicity observed in these patients could be related to the same phenomenon. We focused on three patients who each developed atypical neurotoxicity around 3–4 weeks after cilta-cel infusion (Supplemental Table 3, Fig. 6). All three patients had previously had brief episodes (1–3 days) of grade 1 CRS. At the time of diagnostic lumbar puncture, all three patients showed a much higher percentage of CAR T (Fig. 6A) in the cerebrospinal fluid (CSF) compared to their peripheral blood (PB). In each case, the proportion of CD4+ CAR T was higher in CSF than PB obtained on the same day (Fig. 6B) leading to a markedly skewed CD4/CD8 ratio in the CSF. In both PB and CSF, most CD4+ and CD8+ CAR T consisted of EM-type cells (Fig. 6C). When we measured concentrations of CAR T-related cytokines in the PB and CSF, we found a distinctive cytokine signature in the CNS across all three

patients, with elevated levels of TH1-type and effector cytokines such as MIP-1β, TNF-β, Granzyme B, MCP-1, IL-7, IL-13, and IFN-γ (Fig. 6D).

## Discussion

BCMA-targeted CAR-T cells have revolutionized the treatment of patients with MM. In a recent multi-center study of relapsed-refractory patients with MM who received BCMA-targeted CAR T cell therapy outside of clinical trials, Hansen et al. show that treatment with cilta-cel is associated with better clinical outcomes but also increased rates of high-grade CRS and delayed neurotoxicity (Hansen et al., in revision). However, the immune mechanisms underlying this difference in clinical behavior have remained unclear.

We here provide a detailed view into post-infusion CAR T-cell characteristics in 39 MM patients treated with cilta-cel and ide-cel outside of the controlled clinical trial environment. In this first report of a side-by-side, prospective study in the real-world setting, we find that cilta-cel CAR T show a delayed but more pronounced and prolonged expansion compared to ide-cel. Several associations with clinical outcomes of CAR T cell treatments have been identified, such as CD19-directed CAR T cell dose in patients with B cell lymphoma (BCL)[10,11]. However, in BCL the proliferative capacity of CAR T seems to be even more important[10–20]. Accordingly, in patients with MM we found that higher peak expansion of CAR T, in the case of cilta-cel driven by a pronounced CD4+ CAR T expansion, was positively associated with clinical responses. In agreement with our findings, Fischer et al., observed lower numbers of CAR T in clinical non-responders vs. responders[21], ide-cel expansion as evaluated by qPCR correlated with clinical efficacy[22], and ide-cel recipients showed a strong association of CAR T peak expansion with PFS[23].

Comparing ide-cel to cilta-cel in single patients, we found that higher peak CAR T counts in the cilta-cel group were associated with a switch from the expression of genes associated with cellular proliferation to genes associated with EM T cells. At the same time, after cilta-cel infusion, circulating CAR T cells showed a phenotypic shift toward $T_{EM}$ with lower levels of $T_{SCM}$ compared to ide-cel. When we examined the in vivo characteristics of the expanded cilta-cel CAR T in more detail, we found that compared with CAR- T cells from the same patient, CAR+ T cells showed an overexpression of genes involved in effector function and downregulation of genes indicating regulatory function in CD4+ T cells.

Studies in other hematologic malignancies identified CD4/CD8 double-negative CAR T associated with enhanced CAR T persistence[24]. We did not observe a reliable population of CD4/CD8 double-negative CAR T cells in our patients receiving BCMA CAR T, at least not during the comparably limited follow-up period. However, we did observe a consistent increase in a specific and easily identifiable subpopulation of CAR T, CD4+CD8+ double-positive T cells, following cilta-cel treatment. Some studies have indicated unique tumor-targeting properties of these cells [15] but they also seem to play a role in promoting autoimmune [16] and Human T cell leukemia virus type 1 (HTLV-1)-associated CNS disease [17]. Importantly, a recent study identified a CD4+/CD8+ double-positive T cell population in allo-HCT recipients. In these patients, the presence of CD4+/CD8+ was predictive of ≥ grade 2 GVHD and they were sufficient to mediate xeno-GVHD pathology when retransplanted into naïve mice but provided no survival benefit when mice were challenged with a human B-ALL cell line [18]. Overall, we consider it possible that in our patients this CAR T subpopulation could play a role in mediating anti-myeloma responses but also in promoting off-tumor toxicities.

The higher peak CAR T expansion, and likely also the shift in memory phenotype and gene expression profile, associates with the enhanced persistence of cilta-cel CAR-T over time. As a consequence, up to Month +3, CAR T counts were higher in recipients of cilta-cel vs. ide-cel. In some cilta-cel patients, CAR T were easily detectable by flow cytometry as late as 6 or 12 months post-treatment. In vivo expansion

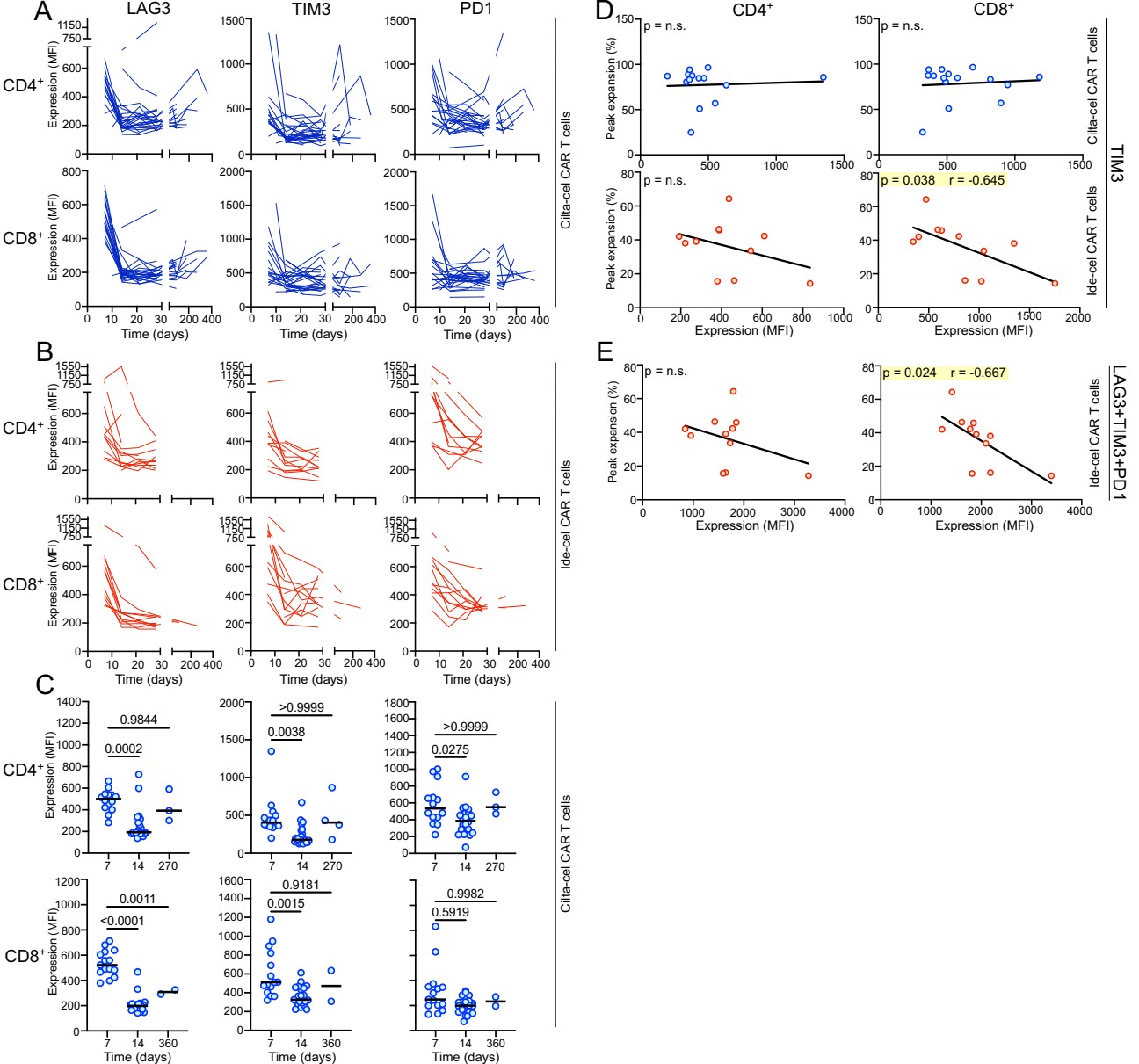

**Fig. 5 | Cilta-cel, but not ide-cel, CAR T show a lack of a short-term correlation between TIM3 expression and proliferation with an upregulation of negative regulatory markers on long-term persisting cells.** To explore potential mechanisms underlying long-term persistence of cilta-cel CAR-T we determined levels of negative regulatory markers LAG3, TIM3, and PD1 on the CAR-T in both patient groups right after infusion into the patient and up to 360 days post-infusion. The highest levels of the negative regulatory markers for both (**A**) cilta-cel (blue) and (**B**) ide-cel (red) were observed on CD4[+] and CD8[+] CAR-T right after infusion followed by a decrease over the next few weeks. **C** Long-term persisting cilta-cel CAR-T showed a second "late" increase in all three negative regulatory markers, often referred to as exhaustion markers, at around one year after CAR-T infusion. Figures show mean fluorescence intensity (MFI) of the given marker on the given CAR-T subtype as measured by flow cytometry over time. The black lines indicate median values and statistical differences between groups were calculated using a two-sided Mann-Whitney U test. **D** Only in the ide-cel group (red), but not in the cilta-cel group (blue), the "early" peak in expression of negative regulatory marker TIM3 predicted a reduced CAR-T in vivo proliferation a few days later. **E** There was an even stronger effect when all three markers were combined. For correlative analyses a Pearson correlation coefficient was calculated. The black line shows the results of a linear regression analysis. Source data are provided as a Source Data file.

of ide-cel vs. cilta-cel has never been compared side-by-side, however, the results of KarMMa-1 seem to support our findings in ide-cel patients demonstrating an initial CAR T peak followed by substantial decline over the subsequent few weeks, with no detectable transgene in most patients by 6 months[2]. Similarly, KarMMa-3 showed very low ide-cel expansion rates and at 12 months most patients had no detectable transgene in their blood[25]. One of very few studies using flow cytometry revealed an initial ide-cel CD3[+] CAR T peak, however, at 3 months most patients had no detectable CAR-T regardless of their

remission status[21]. Overall, we consider it possible that the marked expansion of CD4[+] T-helper CAR T in the cilta-cel contributed to maintaining total CAR T cell numbers over time. Our data indicate that this phenomenon was most likely not based on differences in the T cell starting material but that it is due to a growth advantage of CD4[+] cilta-cel CAR T during manufacturing and after transfer into the patient.

Based on the prolonged expansion and persistence of the cilta-cel cells vs. ida-cel we investigated levels of exhaustion, a main reason for CAR T failure[26–28]. MM patients resistant to BCMA-CAR T manifest a

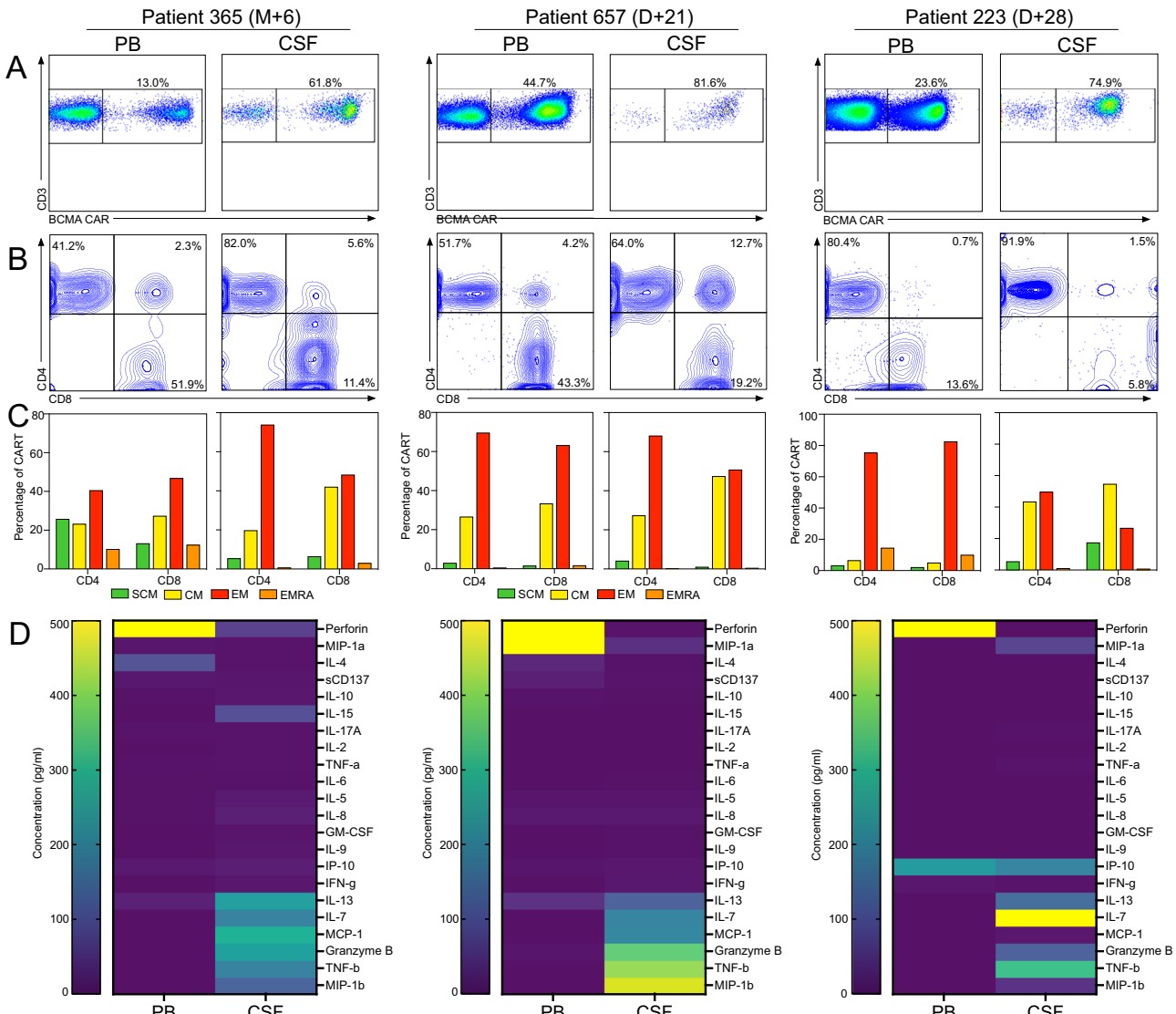

**Fig. 6 | Cilta-cel patients with atypical neurotoxicity show CNS infiltration by effector memory-type CART cells producing an inflammatory cytokine signature. A** Analysis of BCMA-targeted CAR T cell numbers in the peripheral blood (PB) and the cerebrospinal fluid (CSF) from the same timepoint after CART infusion in three myeloma patients who had developed unusual types of neurotoxicity around 3-4 weeks after cilta-cel infusion. Analyses were performed by flow cytometry following co-staining with anti-CD3 and BCMA CAR detection reagent which represents a fluorescent, full-length, recombinant BCMA protein binding to the CAR expressed on the cell surface. **B** CAR T cell subpopulations in the PB and CSF were determined following staining with anti-CD4 and anti-CD8 monoclonal antibodies. **C** Different CAR T memory subpopulations in the PB and CSF were identified by flow cytometry using co-staining with anti-CD45RA and anti-CD62L monoclonal antibodies. **D** The concentration of 22 cytokines/chemokines were measured in the PB and the CSF from the same sample of all three patients using CodePlex Secretome technology. Results are shown as absolute concentrations in pg/mL. Source data are provided as a Source Data file.

tumor microenvironment enriched in terminally-exhausted T cells[29] and CAR T derived from patients with relapsed MM demonstrate reduced in vitro function compared to CAR T manufactured from patients with newly diagnosed disease[30]. In our patients with MM, regardless of CAR T product, we found the highest proportion of CAR T cells expressing negative regulatory markers right after product infusion, with proportions subsequently decreasing over the next few weeks. Long-term persistence of CAR T cells was almost exclusively identified in the cilta-cel recipients; these recipients' CAR T cells showed a second "late" peak in the expression of negative regulatory markers - one year after CAR T infusion. Importantly, we found only in the ide-cel group that the "early" peak in expression of the negative regulatory marker TIM3, or of all three negative regulatory markers combined, predicted reduced in-vivo CAR T proliferation a few days

later. There was no such association in the cilta-cel group indicating a difference in the regulation of cellular proliferation in these cells.

Aiming at identifying factors involved in the protection of cilta-cel cells from exhaustion we investigated CD127 on the CAR T over time. Expression of CD127 (IL-7Rα) is crucial for T cell differentiation and survival[7] and transgenic expression of its ligand IL-7 has been shown to prevent CAR T exhaustion and enhance persistence[9]. Furthermore, IL-7 fused with hybrid Fc (rhIL-7-hyFc) promotes proliferation, persistence and cytotoxicity of human CAR T in preclinical models[31]. Importantly, we found that the long-term persistent cilta-cel CAR-T had markedly upregulated expression of CD127 with a statistically significant upregulation specifically in CD4[+] CAR T which, although it may also represent a reduction in CAR T with a CD127[low] T regulatory phenotype[32], may confer a survival advantage on the cilta-cel cells.

Cell surface expression of CD27 may be another factor contributing to the long-term survival of the cilta-cel cells and we found that even early after CAR T infusion, patients who had received cilta-cel showed much higher levels of CD27 on their CD4[+] and CD8[+] CAR T than ide-cel patients. This difference was maintained through the first 4 weeks, and in both patient groups there was a significant positive correlation between CD8[+] CAR T peak expansion and CAR T expression of CD27. Accordingly, in BCL patients the presence of a population of CD27[+]PD-1[-]CD8[+] CAR T predicted clinical responses[33], and CAR T transduced with a CD27 costimulatory domain showed increased antigen-stimulated effector functions[34]. CD27-mediated signaling may promote memory- rather than effector-associated gene programs leading to improved tumor control[35], consistent with findings that signaling through the CD70-CD27 axis may improve the efficacy and persistence of CAR T[4].

Both ide-cel and cilta-cel are based on second-generation CAR scaffolds using a CD8α hinge domain and 4-1BB and CD3ζ signaling domains, suggesting that the functional differences we observed derive predominantly from the constructs' respective antigen-binding domains. Generating the respective CAR T subtypes in vitro, we observed, similar to the in vivo situation, that cilta-cel CAR T showed stronger cell-surface expression of the CAR and a higher CD4/CD8 ratio. In addition, cilta-cel effector/memory CAR T spontaneously, and in the absence of target antigen, secreted IL-2 and TNF-α and demonstrated a significantly greater mitochondrial metabolic capacity.

Our clinical observations show a higher infection rate as well as a higher rate of atypical neurotoxicity in the cilta-cel group relative to the ide-cel product. These are in agreement with observations from the CARTITUDE-1 and CARTITUDE-4 trials[3,36]. The distinct in-vivo and in-vitro characteristics of the products as well as differences in their clinical activity and toxicity profile prompted a detailed look into the characteristics of cilta-cel CAR T infiltrating the CNS in our patients with neurotoxicity. Neurotoxicity including its most common form - ICANS - is a potential side effect of CAR-T therapies[37]. The exact cause of CAR T-mediated neurotoxicity is not fully understood, but it is thought to be related to cytokine release by CAR-T both within the CNS and in the periphery. In addition to ICANS, other, atypical CAR T-related neurotoxicities, such as movement disorders, cognitive impairment, and personality changes, have been described[37]. The pathophysiology of atypical neurotoxicity is even less well understood.

When we analyzed PB and CNS-infiltrating CAR T in our patients with atypical cilta-cel-induced neurotoxicity, we found a proportional enrichment of CAR T in the CNS with a predominance of CD4[+] T cells showing an EM phenotype. These CD4[+] CAR T were characterized by a distinct gene signature and promoted the secretion of Th1-type and effector cytokines such as MIP-1β, TNF-β, Granzyme B, MCP-1, IL-7, IL-13, and IFN-γ. We consider it possible that this distinct subpopulation of effector-type CD4[+] CAR T infiltrating the CNS is at least partially responsible for the observed neurotoxicity. Our in vitro experiments showing that the cells are capable of constitutively producing these cytokines in the absence of antigen contact make such a scenario even more likely. While our work does not conclusively identify the molecular mechanism driving cilta-cel's unique pharmacokinetics, prior work demonstrated efficient CAR multimerization and formation of a more robust immune synapse by the biparatopic cilta-cel[38]. In addition, it is conceivable that distinct biophysical properties of this class of CARs or at least this specific construct may drive some of the observed behavior[39].

This is the first report of an in-depth, prospective, comparative analysis of immune responses to treatment with cilta-cel vs ide-cel in the real-world setting; herein, we have correlated our findings with clinical responses and toxicities. In total, our combined findings indicate that the cilta-cel product's intrinsic characteristics, such as higher CAR surface expression, less of an impact of an "early" upregulation of

negative regulatory factors, and constitutive antigen-independent activation, result in an enhanced initial expansion of the cells followed by increased persistence. These intrinsic biological differences may partially explain the differences in clinical activity and toxicity profiles observed in myeloma patients in the real-world setting. Additional work is needed to better understand the molecular causes for the observed differences in expansion, cytokine production, and phenotype, as well as the pathomechanisms driving atypical neurotoxicity in a subgroup of patients.

## Methods

### Patient samples
Patients with MM scheduled for treatment with CAR T were consecutively enrolled and samples and clinical data were collected under Institutional Review Board (IRB)-approved protocol 2043GCCC (IRB HP-00091736) as released by the University of Maryland Baltimore for the immunomonitoring of lymphoma/myeloma patients following CAR T treatment. All participants provided written informed consent and no participant compensation was provided. Sex and gender were considered as part of data analysis.

### Flow cytometry
Prior to analysis of stained patient samples by flow cytometry, compensation settings were determined using single color controls and unstained cells. Single color controls were prepared using the MACS Comp Bead kit (Miltenyi, cat. no. 130-104-693). Cells were washed and stained in PBS containing 2% bovine serum albumin (BSA). Staining of PBMCs or cell pellets generated from cerebrospinal fluid (CSF) was performed using a panel of monoclonal antibodies and CAR Detection Reagents (Supplemental Table 1) following the manufacturer's instructions. Live cells were identified by 7-AAD dye exclusion (Miltenyi, cat. no. 130-111-568). Samples were acquired using a Miltenyi MACSQuant Analyzer 10.

### CodePlex secretome analysis
Cytokine/chemokine concentrations in CSF and plasma samples were quantified using the CodePlex Secretome Human Adaptive Immune Panel kit (IsoPlexis, cat. no. CODEPLEX-2L01). This panel measures the absolute concentration of 22 cytokines in a single sample using internal cytokine standards. To carry out the CodePlex analysis, chips were thawed at room temperature for 1 hour before supernatants were loaded onto the chip microchamber. The chip was then loaded into the Isolight reader (Isoplexis, Branfold, CT) and automated analysis of raw data was performed using IsoSpeak software (Isoplexis).

### Data analysis
Flow cytometry data were analyzed using FlowJo software version 10.9.0 (BD Biosciences, Franklin Lakes, NJ). Overall data analysis was performed using GraphPad Prism software version 9.5.1 (GraphPad Software, Boston, MA). Figures were composed using OmniGraffle software version 7.21.4 (The Omni Group, Seattle, WA).

### Reporting summary
Further information on research design is available in the Nature Portfolio Reporting Summary linked to this article.

## Data availability
The original data underlying Figs. 1–6 are provided as a Source Data file. All additional data in the article or Supplementary Data are available from the corresponding author upon request to facilitate a more meaningful and selective exchange of relevant data. Individual de-identified participant data will also be shared upon request in agreement with the informed consent provided by the patients. Source data are provided as a supplement to this paper. The raw RNA Sequencing data used in this study are publicly available in the NCBI/

SRA database under BioProject ID PRJNA126199 (accession codes GSM9004223, GSM9004224, GSM9004225, GSM9004226 GSM9004227) using the following link: https://www.ncbi.nlm.nih.gov/bioproject/?term=PRJNA1261997. Gene Expression Omnibus (GEO) data are completed and available online under the following GEO submission links: https://www.ncbi.nlm.nih.gov/geo/query/acc.cgi?acc=GSE298011. https://www.ncbi.nlm.nih.gov/geo/query/acc.cgi?acc=GSE298012. Data Sharing Plan: A. Data to be shared will include de-identified participant data; B. These data will include clinicopathological data, immunohistochemistry data, and additional flow cytometry data; C. These data will become available upon publication of this article and will be available for 10 years; data will potentially be shared electronically, i.e. for subgroup or meta-analyses, with any type of non-profit institution. Source data are provided with this paper.

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

## Acknowledgements

This study was funded by two grants from the Kahlert Foundation (to D.A.), by the Maryland Department of Health's Cigarette Restitution Fund Program (to D.A. and X.F.) and by the National Cancer Institute - Cancer Center Support Grant (CCSG) P30CA134274. In addition, the D.A. lab received philanthropic support from the Becker Family Foundation, Marco Chacon and the Kassap Family Foundation.

## Author contributions

D.A. designed the study, performed experiments, analyzed the data, made figures, and wrote the manuscript. T.L. analyzed the data, prepared figures, and wrote the manuscript. P.L. collected and processed patient samples and clinical data. D.O., D.Y., A.M., E.G. and R.M. processed patient samples and performed experiments. D.S., P.H. and X.W. designed experiments, performed experiments, analyzed data, and wrote the manuscript. A.C.S., L.T., R.K. and M.E.K. performed experiments, analyzed data, and wrote the manuscript. I.P. and R.S. collected clinical data and wrote the manuscript. X.F., K.A.D., A.A.F., K.G.H., S.D., J.A.Y., A.B., N.M.H. and A.P.R. analyzed data and wrote the manuscript. M.H.K. collected clinical data, analyzed data, and wrote the manuscript.

## Competing interests

T.L. receives a salary from AbbVie. The other authors declare that they do not have any competing interests.

## Additional information

**Djordje Atanackovic** [1,2,3,4] ✉, **Tim Luetkens** [1,3,4], **Dina Schneider** [5], **Peirong Hu**[5], **Xu Wang**[5], **Amol C. Shetty** [6], **Luke Tallon**[6], **Imari Patel**[1,2], **Rohan Singh**[1,2], **Etse Gebru**[1,3], **Rediet Mulatu**[1,3], **Destiny Omili**[1,3], **Daniel Yamoah**[1,3], **Xiaoxuan Fan** [1,4], **Aerielle Matsangos**[1,3], **Patricia Lesho**[1], **Kenneth A. Dietze** [4], **Ariel A. Fromowitz**[1,2,3], **Kim G. Hankey** [1,2,3], **Saurabh Dahiya**[7], **Jean A. Yared**[1,2,3], **Nancy M. Hardy**[1,2,3], **Rima Koka**[8], **Michael E. Kallen**[8], **Ashraf Badros**[1,2,3,9], **Aaron P. Rapoport**[1,2,3,9] & **Mehmet H. Kocoglu**[1,2,3,9]

[1]University of Maryland Greenebaum Comprehensive Cancer Center, Baltimore, MD, USA. [2]Department of Medicine, University of Maryland School of Medicine, Baltimore, MD, USA. [3]Transplant and Cellular Therapy Program, University of Maryland Greenebaum Comprehensive Cancer Center, Baltimore, MD, USA. [4]Department of Microbiology and Immunology, University of Maryland, Baltimore, MD, USA. [5]Lentigen Technology Inc., a Miltenyi Biotec Company, Gaithersburg, MD, USA. [6]Institute for Genome Sciences (IGS), School of Medicine, University of Maryland Baltimore, Baltimore, MD, USA. [7]Stanford University, Stanford, CA, USA. [8]Department of Pathology, University of Maryland School of Medicine, Baltimore, MD, USA. [9]These authors jointly supervised this work: Ashraf Badros, Aaron P. Rapoport, Mehmet H. Kocoglu. ✉e-mail: datanackovic@som.umaryland.edu

