## [Transparent Peer Review file · Nature Communications]

Immune correlates of anti-BCMA CAR-T products idecabtagene vicleucel and ciltacabtagene autoleucel in a real-world cohort of patients with multiple myeloma

Corresponding Author: Professor Djordje Atanackovic

Version 0:

Reviewer comments:

Reviewer #1

(Remarks to the Author)

Dr. Atanackovic et al. compared the CAR-T kinetics, immunophenotypes, and functionalities of two approved anti-BCMA CAR-T cell products, Cilta-cel and Ide-cel, in a real-world setting. The two commercial agents feature distinct antigen-recognition domains, which may lead to different mechanisms of action against multiple myeloma. The laboratory data, which were based on clinical efficacy and toxicity, showed how the two products worked by showing the dynamic changes of CAR-T in patients from different angles. The study presents significant results that may yield the optimized design ideas for future CAR-T cell manufacturing.

Here are a few comments:

1. Table 1: It's known that baseline tumor burden and lymphodepletion conditioning scheme may have an impact on CAR-T cell expansion. Can the authors include the details?
2. Figure 1: The authors described the post-treatment CD4 and CD8 CAR-T cell dynamics. What are the CD4 and CD8 compositions in the infused products, as well as memory phenotypes? Is the CD4/CD8 composition of the two infused CAR-T products different than that of the patients' host T cells?
3. Supplementary Figure 2: It's interesting that the Cilta-cel group contained a subset of 'atypical' lymphocytes. Are they CAR-T cells?
4. Supplementary Figure 3E: Can the correlations of peak counts and response depth be performed in the Cilta-cel and Ide-cel, separately?
5. The study showed that Cilta-cel exhibited an effector-memory phenotype at peak expansion. Is this phenotype associated with a high production of IL-2 and TNF- α demonstrated by ex vivo assay?
6. The presence of anti-drug antibody (ADA) affects CAR-T's persistence. Cilta-cel and Ide-cel both have non-human fragments. Have the authors detected ADA?
7. Cilta-cel has two antigen-binding heavy chains. High-quality synapse between Cilta-cel and tumor cell leads to enhanced cytotoxicity (Sun Y et al., Signal Transduction and Targeted Therapy, 2024). Could this structure be linked to Cilta-cel's distinct CAR-T kinetics and cytokine release compared to Ide-cel?
8. Figure 6: Three patients who suffered non-ICANS neurotoxicity had CAR-T cells detectable in CSF. Did the three patients develop constitutional CRS? Have the authors detected the CAR-T cells and cytokines in the CSF of the patients with ICANS? If available, are there any differences in CSF CAR-T immunophenotypes and cytokine profiles between ICANS neurotoxicity and non-ICANS neurotoxicity?

Reviewer #2

(Remarks to the Author)

In this manuscript, Atanackovic et al evaluate CAR T-cell clinical outcomes and immune dynamics in a single-center cohort of real-world multiple myeloma patients receiving cilta-cel or ide-cel, with a goal of better understanding how immune dynamics associate with treatment efficacy and toxicity. Consistent with other studies, the authors report higher response rates for cilta-cel vs ide cel, albeit non-statistically significant. Using longitudinal peripheral blood samples from a 39-patient cohort, the authors assessed circulating CAR T-cell persistence and identified differences in peak CAR T-cell expansion between cilta-cel and ide-cel patients, as well as differential patterns of T-cell subsets between the two CAR T-cell products. Although the manuscript addresses an important and timely topic, the clinical findings are of limited novelty, and the immune dynamics interpretations are not sufficiently supported by the data.

Major comments:

1. For clinical outcomes, the cilta cel patients showed non-statistically higher ORRs at 1 and 3-months than idel cel patients. In a non-clinical trial setting, several confounding variables could skew responses. For example, a much higher fraction of cilta cel patients (78.3%) vs ide cel patients (43.8%) received bridging therapy before CAR T cell treatment. What was the disease status, including depth of response, for the cilta cel vs idel cel groups before CART infusion? Was a subgroup analysis comparing patients who received bridging therapy vs not performed? Also, what was the year range for the CAR T treatments, as this could also have a bearing on the choice/availability of therapy, use of bridging, etc?
2. The high rate of infections (78.3%), especially bacterial (60.9%), reported for the cilta-cel patients is a bit surprising. If the infections occurred early after CAR T cell infusion, a product-specific cause would seem unlikely. When did the infections occur? Were the bacteria identified? Also, what was the nature of the infections – pneumonia, bacteremia, catheter-associated, etc?
3. The authors note more double-positive CD4/CD8 T cells post-cilta cel (Figure 1). However, these cells constitute a very minor fraction of total lymphocytes and are of unclear clinical relevance. Of note, studies in other hematologic malignancies have identified CD4/CD8 double-negative CAR T cells (Anderson ND et al Nat Med 2023) associated with CAR T-cell persistence. Were double-negative populations observed in this dataset? Also, the relevance of CD127 expression changes (Supplemental Figure 8) is unclear. Due to the limited number of colors of the flow panel used, this may represent a decrease in CAR T cells with a Treg-like phenotype given the global reduction in CD127 expression.
4. The scRNAseq data in Figure 4 is from a single cilta cel patient. How/why was this patient chosen? This is not trivial, as factors such as prior therapies, depth of response, and age, can impact the findings. Of note, the scRNAseq data from Figure 4 demonstrate a significant increase in the CD8 EM population (red bar) when comparing panel C to panels D or E, but the change in the CD8 EM population was not statistically significant in Figure S4. This discrepancy makes interpretation of the data in Figure 4 problematic, as it is representative of a single patient that does not align with the trend of the cilta cel cohort. Also, a comparison with a matched ide cel patient(s) would be interesting/insightful.
5. The authors state in the abstract that “comparably longer persistence of cilta cell CAR T was associated with increased resistance to exhaustion...” and subsequently in the results state “Remarkably, we found the highest levels of exhaustion markers in cilta cel and ide cel-treated patients across T cell subsets immediately after infusion. This initial upregulation was followed by a decrease in exhaustion markers over the following weeks.” This observation is not indicative of exhaustion, as transiently increased expression of negative regulatory markers (PD-1, LAGE-3, etc) is expected in the setting of T-cell activation in the early window after CAR T-cell infusion. Also, this observation does not signify *resistance* to exhaustion.

Minor comments:

1. The specific time points for sample collections should be included in the methods and/or results section.
2. Additional details regarding methods of flow-cytometric-based analysis of CAR T-cell populations are needed (can be described in the supplement). While the authors provide information about anti-CAR antibodies in supplemental Table 1, the method of CAR T-cell identification is not shown beyond what was shown in Figure 3A. Were FMOs used or was the gate in Figure 3A drawn arbitrarily based on the fluorescence intensity histogram? The use of FMO on a population gated for this type of analysis is essential to accurately discriminate positive and negative cell populations.
3. Comparisons between duration of response or total cohort progression-free survival should be included in Table 1, and not just overall survival in supplemental Figure 3. This is especially important to inform the analysis of the differential rates of toxicities between the two therapies that the authors report.
4. Consider an alternative color to the green used in Figures 2D, S3B, S3E, S5A, and S5C, as when contrasted with adjacent similar greens and reds (see Figure S5), may prove difficult to perceive by readers with deuteranopia.
5. Supplemental Figure 1: a) Labeled as “Overexpression” of CD27.... “Higher” or “increased” would be more accurate/appropriate. b) The inclusion of a representative dot plot with gating is recommended.
6. References: #5 & 6 need to be reformatted

Reviewer #3

(Remarks to the Author)

Distinct Expansion, Phenotype, Function, and Toxicity of Cilta-Cel vs. Ide-Cel CAR T in the Real World

Relapsed multiple myeloma remains a significant therapeutic challenge, but immunotherapy with CAR T cells has considerably advanced the care of these patients.

This manuscript explores key differences between Ide-Cel (Idecabtagene vicleucel) and Cilta-Cel (Ciltacabtagene autoleucel) in terms of initial expansion at various time intervals, phenotypic differences, and their association with functional/clinical outcomes. The study also highlights the unique toxicity profiles of both CAR T therapies. Notably, the authors investigate the occurrence of central nervous system (CNS) toxicity, with cerebrospinal fluid (CSF) and peripheral blood (PB) testing revealing novel findings. Additionally, the research examines a unique feature of atypical cells in the post-CAR T period, providing detailed morphological and immunophenotypic definitions. Another finding was the distinctive difference in the longer persisting CAR T cells and unique phenotypic findings.

Findings from this real-world study present noteworthy results that could lead to improved management and care for these patients. I approve this manuscript for publication without further edits.

Reviewer #4

(Remarks to the Author)

Version 1:

Reviewer comments:

Reviewer #1

(Remarks to the Author)

The authors have addressed all of my concerns.

Reviewer #2

(Remarks to the Author)

The authors have addressed most of the comments, but the following issues require attention:

1) In the Abstract, the authors write: "The pronounced expansion of cilta-cel CAR T was associated with higher CAR and CD27 expression and reduced impact of upregulation of negative regulatory factor TIM3 on the infused cells." ==> The data show a "decrease" in TIM-3 expression, but conclusions about the functional significance or "impact" of this observation are speculative. This should be revised to ensure the findings are not overstated.

2) In the Discussion (page 16, 2nd paragraph), the authors write: "Importantly, we found only in the ide-cel group that the "early" peak in expression of the exhaustion marker TIM3, or of all three markers combined, predicted reduced in-vivo CAR T proliferation a few days later. There was no such association in the cilta-cel group indicating that the freshly produced cilta-cel cells may be more resistant to the immediate impact of negative regulatory factor TIM3." ==> Similar to comment #1, this section should be revised as TIM-3 expression alone is not a definitive marker of exhaustion. Additionally, it is not accurate to conclude that cilta-cel cells are "resistant" to the "immediate impact of the negative regulatory factor TIM-3." This statement overinterprets the data and should be reworded for accuracy.

3) In the Discussion (page 17, 1st paragraph, last sentence), the authors write: "...although it may also represent a reduction in CAR T with a CD127^{low} T regulatory phenotype, most likely confers a survival advantage on the cilta-cel cells. ==> "... most likely confers a survival advantage..." needs to be revised to "may confer a survival advantage."

4) In the Figure Legend 5 (page 27), the title of "Cilta-cel CAR-T show short-term resistance to exhaustion with an upregulation of exhaustion markers on long-term persisting cells" needs to be revised to remove "resistance to exhaustion" and change "exhaustion markers" to "negative regulatory markers."

5) In the Figure Legend 5 (page 27): The multiple references to "exhaustion markers" should be changed to "negative regulatory markers."

6) Results section page 12 on "Long-term persistence...with upregulation of exhaustion markers and IL-7Ra." ==> The phrase "exhaustion markers" is inaccurate, as transiently increased expression of negative regulatory markers (e.g., PD-1,

LAG-3) is an expected response to T-cell activation in the early period following CAR T-cell infusion. Recommend rewording this section, as the authors have not demonstrated T-cell exhaustion and may be overstating their findings.

7) Response to major comment #4: Presenting this data in a main figure is questionable and potentially misleading, given that it is based on a single patient. As previously noted, this raises substantial concerns about its limitations and lack of generalizability.

Reviewer #4

(Remarks to the Author)

Version 2:

Reviewer comments:

Reviewer #2

(Remarks to the Author)

The authors have addressed most of the comments, but the following issues require attention:

1) Results section page 10, 2nd paragraph: "To substantiate the phenotypic change, we characterized ciltacicel CAR T over time using single-cell RNA sequencing on one of the first ciltacicel patients enrolled in our study showing the typical massive increase in effector-type CAR T." ==> The word "massive" is hyperbolic and should be deleted. Also, "increase" is written twice.

2) Results section, page 12: "Long-term persistence...with upregulation of exhaustion markers and IL-7Ra." (Comment #6 from the previous review):

a) The phrase "exhaustion markers" is still present in the title of this section and should be revised to "negative regulatory markers."

b) Second sentence: "This initial upregulation was followed by a decrease in these surface molecules, often referred to as exhaustion markers, over the following weeks..." ==> The words "often referred to as exhaustion markers" should be deleted, as this implies that the markers alone are indicative of exhaustion, which they are not.

c) Second paragraph, first sentence: "In addition to reduced exhaustion, response to pro-proliferative soluble factors may contribute to long-term CAR-T persistence." ==> "Reduced exhaustion" has not been definitively shown, so "In addition to reduced exhaustion" should be deleted/rephrased.

3) Discussion, second paragraph, page 16: In sentences 3 and 4, the authors again refer to exhaustion, which they have not shown. The phrase "exhaustion markers" is inaccurate, as transiently increased expression of negative regulatory markers (e.g., PD-1, LAG-3) is an expected response to T-cell activation in the early period following CAR T-cell infusion. Need to rephrase to "negative regulatory markers."

As requested, please find below our point-by-point response to all comments:

REVIEWER 1

Comment #1: “Table 1: It’s known that baseline tumor burden and lymphodepletion conditioning scheme may have an impact on CAR-T cell expansion. Can the authors include the details?”

Thank you very much for this comment. In the revised version of the manuscript (Methods Section) we are now explaining that all patients in both cohorts received lymphodepleting chemotherapy and all but one single patient received the standard fludarabine and cyclophosphamide regimen for three days. The fludarabine was dose-adjusted for renal function. One patient received a bendamustine-based regimen during the time of fludarabine shortage in the US.

We agree with the reviewer that the tumor burden at baseline can have an impact on CAR T cell expansion. Historically, in clinical trials, bone marrow plasma cell involvement as well as serum beta-2 microglobulin (b2m) levels have been used as surrogates of tumor burden. Unfortunately, in our real-world study, there was a significant number of patients did not have bone marrow biopsies prior to lymphodepletion. However, in response to the reviewer’s comment, we obtained available pretreatment b2m levels and, as explained in the revised version of the manuscript, we found no significant difference between cohorts in b2m levels at baseline. Median serum b2m levels were 2.7 mg/L [95%CI 2.3-3.9] in the cilta-cel group and 2.8 mg/L [95%CI 2.1-4.3] in the ide-cel group.

Comment #2: “Figure 1: The authors described the post-treatment CD4 and CD8 CAR-T cell dynamics. What are the CD4 and CD8 compositions in the infused products, as well as memory phenotypes? Is the CD4/CD8 composition of the two infused CAR-T products different than that of the patients’ host T cells?”

This is an excellent and very important point. Unfortunately, our cellular therapy Standard Operating Procedure (SOP) requires a post-infusion bag wash which is then infused to the patient. Consequently, we do not have access to the infused product for analysis, not even to small-volume leftovers from the procedure. However, in response to the reviewer’s much appreciated comment we performed an extensive literature search and we found out that in clinical studies such as the phase I trial CRB-4-1, the composition of the ide-cel product was highly variable containing a median of 85% of CD4⁺ and 13% CD8⁺ CAR T cells [1].

With respect to cilta-cel, in the CARTITUDE-1 study, the product contained a mixture of transduced and non-transduced T cells; the median transduction efficiency was 16% (range 5-32%), with a balanced distribution of CAR⁺CD4⁺ and CAR⁺CD8⁺ cells with a median frequency 12% (range 2-28%) and 6% (range 2-20%), respectively [2].

As an additional response to this reviewer's comment, we have reanalyzed our data from the time of apheresis and we found that when the starting material for CAR T production was collected, there was no difference within either the ide-cel or cilta-cel group in terms of peripheral blood CD4⁺ and CD8⁺ Non-CAR T cells. Using these results, we have composed a new Figure 1E. In the revised version of the manuscript, we are now explaining that the marked expansion of CD4⁺ T cells is probably not based on differences in the starting material collected from the patient but that it is most likely due to the enhanced expansion of CD4⁺ CAR T among cilta-cel T cells during manufacturing (as shown in Supplemental Figure 6H) and after reinfusion into the patient. We have also amended the Results and Discussion sections to reflect these additional findings.

Comment #3: “Supplementary Figure 2: It’s interesting that the Cilta-cel group contained a subset of ‘atypical’ lymphocytes. Are they CAR-T cells?”

We agree with the reviewer that this is a very interesting (and reproducible) phenomenon. These lymphocytes, which look “atypical” by morphology, are indeed CAR T cells as confirmed by flow cytometry. This is now explained in the revised version of the manuscript and in the figure legend for Supplementary Figure 2.

Comment #4: “Supplementary Figure 3E: Can the correlations of peak counts and response depth be performed in the Cilta-cel and Ide-cel, separately?”

Thank you for this excellent comment. We had indeed looked for associations between peak CAR T cell counts with the depth of the responses separately for each patient group and we did not find any significant correlations. We think that this negative finding was most likely due to relatively homogenous types of responses within each group and small group sizes. This is now explained in the revised version of the manuscript.

Comment #5: “The study showed that Cilta-cel exhibited an effector-memory phenotype at peak expansion. Is this phenotype associated with a high production of IL-2 and TNF-α demonstrated by ex vivo assay?”

These are very important points and we can confirm that the effector memory phenotype in cilta-cel patients at peak expansion is indeed concordant with the capability of cilta-cel T cells for enhanced production of IL-2 and TNF-α *ex vivo*. In response to the reviewer's comment, we have actually performed additional experiments demonstrating that the cilta-cel CAR T cells did indeed express the phenotype of effector-memory T cells at the same timepoint during cell culture when a significantly enhanced production of IL-2 and TNF-α was observed (see revised and amended Supplemental Figure 6). We have also amended the Results and Discussion sections to reflect these additional findings.

We would also like to confirm that, as implicated by the reviewer, IL-2 is indeed important for both T cell effector differentiation and T cell memory formation. Specifically, autocrine

IL-2 production by CD8⁺ stem-like memory T cells promotes survival, preservation of memory phenotype, resistance to exhaustion, and is required for potent recall responses [3, 4].

TNF- α is produced by activated T cells and can drive activation and proliferation of both naïve and effector T cells through TNFR11 acting as T cell co-receptor [5-7]. Specifically, TNF- α is known to augment levels of IL-2R, and increase cytokine production by T cells [8, 9]. It is therefore plausible that heightened TNF- α secretion by cilta-cel CAR T contributes to greater overall CAR T cell proliferation and IL-2 production, and accumulation of a greater number of effector-memory cells at expansion peak, as observed in our cilta-cel patients.

Comment #6: “The presence of anti-drug antibody (ADA) affects CAR-T’s persistence. Cilta-cel and Ide-cel both have non-human fragments. Have the authors detected ADA?”

We agree with the reviewer that the presence of anti-drug antibodies (ADAs) can potentially affect the persistence and efficacy of CAR T cells. Unfortunately, we were not able to measure ADAs in our patient samples due to the lack of commercially available testing kits. However, in response to the reviewer’s comment we performed an extensive literature search and we found that Sections 6.2 of the package inserts issued by the FDA for both products contained data on the immunogenicity of the CAR constructs.

The immunogenicity of the ide-cel product was measured by assessing levels of anti-CAR antibodies and according to the clinical studies 2.6% (9/349) of patients were positive prior to infusion and 53% (186/349) of patients developed antibodies at some point after receiving the treatment. Importantly, the occurrence of ADA did not have an impact on the cell expansion, safety or efficacy of the ide-cel treatment [10].

The immunogenicity of cilta-cel was measured by assessing antibodies against the extracellular portion of anti-BCMA CAR using a validated assay. In the CARTITUDE-1 study, 19.6% (19/97 patients) became positive for anti-product antibodies whereas in the CARTITUDE-4 study, 21% (39/186 patients) developed anti-CAR antibodies [11].

In addition, for CARTITUDE-2 which included patients with 1-3 prior lines of therapy, overall incidence of ADAs against cilta-cel was 25% for Cohorts A and C and 42% for Cohort B with onset time ranging from 57 to 186 days across 3 cohorts. The authors concluded that this late onset of ADA appearance would argue against any impact on cellular expansion kinetics [12].

Our conclusion from these combined data would be that there is no substantially increased incidence of ADA in ide-cel vs. cilta-cel patients and that in those ide-cel patients who developed ADAs this type of therapy-induced immunity did not impact the expansion of the CAR T cells. Therefore, we would say that the occurrence of ADAs is unlikely to explain the differences observed in our study in terms of expansion and/or persistence of both products. We have amended our manuscript accordingly.

Comment #7: “Cilta-cel has two antigen-binding heavy chains. High-quality synapse between Cilta-cel and tumor cell leads to enhanced cytotoxicity (Sun Y et al., Signal Transduction and Targeted Therapy, 2024). Could this structure be linked to Cilta-cel's distinct CAR-T kinetics and cytokine release compared to Ide-cel?”

We thank the reviewer for this insightful comment. Ciltacabtagene is the first and only approved biparatopic antibody-based therapy to date and it is indeed possible that the relatively uncommon biparatopic ectodomain structure of cilta-cel plays a role in the product's distinct pharmacokinetics. In addition to the mechanism proposed by the reviewer and previously exemplified by Sun et al. based on the formation of a more robust immune synapse between CAR T cells and BCMA-expressing target cells, it is also possible that distinct biophysical properties of this CAR design class or at least this specific construct, cause some of the unique behavior of cilta-cel. Biparatopic structures are postulated to have unique advantages over their monoparatopic counterparts among which are enhanced avidity, receptor clustering and slower dissociation as well as other non-cytotoxic killing related mechanisms of action such as inhibition of multiple intracellular pathways, rapid target protein internalization, some of which may be playing additional roles aside from cell mediated cytotoxic killing and overcome resistance. In response to this comment, we revised our discussion to reflect these possibilities.

Comment #8: “Three patients who suffered non-ICANS neurotoxicity had CAR-T cells detectable in CSF. Did the three patients develop constitutional CRS? Have the authors detected the CAR-T cells and cytokines in the CSF of the patients with ICANS? If available, are there any differences in CSF CAR-T immunophenotypes and cytokine profiles between ICANS neurotoxicity and non-ICANS neurotoxicity?”

These are all very important questions. We can confirm that all three patients had shown prior CRS, however, the maximum degree of CRS was only grade 1 and symptoms lasted for only 1-3 days. This is now explained in the revised version of the manuscript.

We agree that the second question is very important and of substantial clinical relevance. Unfortunately, at this point we are not able to answer the question whether there are elevated levels of cytokines and/or CAR T cells in the CSF of patients with classical ICANS or whether there are any differences in CSF CAR-T immunophenotypes and cytokine profiles between ICANS neurotoxicity and non-ICANS neurotoxicity. The reason for this is simply that diagnostic lumbar punctures are not performed on a routine basis in patients with classical ICANS at our institution. Therefore, we do not have the patient samples needed to address this particular question. We hope that in the future studies will be set up across institutions that will specifically aim at looking into this important question.

REVIEWER 2

Major Comments

Major Comment #1: “For clinical outcomes, the cilta cel patients showed non-statistically higher ORRs at 1 and 3-months than idel cel patients. In a non-clinical

trial setting, several confounding variables could skew responses. For example, a much higher fraction of cilta cel patients (78.3%) vs ide cel patients (43.8%) received bridging therapy before CAR T cell treatment. What was the disease status, including depth of response, for the cilta cel vs idel cel groups before CART infusion? Was a subgroup analysis comparing patients who received bridging therapy vs not performed? Also, what was the year range for the CAR T treatments, as this could also have a bearing on the choice/availability of therapy, use of bridging, etc.?”

We consider these questions very important and we agree with the reviewer that additional information on these points would be very useful. We will address them individually point-by-point.

Proportion of patients receiving bridging therapy: In our study, about three quarters of cilta-cel patients received bridging therapy (76%; 19/25) whereas one quarter was able to receive the cell product without bridging (24%; 6/25). This is very much in line with the data from US Multiple Myeloma Immunotherapy Consortium, which our site is participating in, looking at clinical data from patients receiving cilta-cel in the real-world setting across 16 academic sites. In that study, rates of bridging therapy were 77% (195/255 apheresed patients) and 78% (184/236 infused patients), respectively [13].

Post-bridging clinical responses: In our cilta-cel patients, the depth of responses with various bridging strategies were mostly poor (5 PD, 4 SD, 1 MR). Objective responses were seen in only 20% of the cilta-cel patients (4 PR, 1 CR). The Consortium study reported objective responses of 26% and 27% in the apheresed and infused patients, respectively, which was slightly better. This could indicate slightly more aggressive/refractory disease in our cilta-cel cohorts and might have affected objective response rates. In response to the reviewer’s comment, we performed a subgroup analysis comparing patients who received bridging therapy vs. no bridging therapy. There was no significant difference in terms of response rates to CAR T treatment between patients who had received bridging therapy versus those who had not. This is again very much in line with the data from US Multiple Myeloma Immunotherapy Consortium, which suggested that in cilta-cel patients the response to bridging did not have an impact on PFS [13]. We have updated the Results section of our revised manuscript to reflect the new findings outlined above.

Time of CAR T application: Aphereses in our cilta-cel patients took place between May 2022 and December 2023. This is now explained in the Methods section of our revised manuscript. It certainly is possible that that post-COVID manufacturing slot availability might have had an impact on the choice of product which unfortunately represents a limitation of customized treatments such as autologous CAR T cells.

Major Comment #2: “The high rate of infections (78.3%), especially bacterial (60.9%), reported for the cilta-cel patients is a bit surprising. If the infections occurred early after CAR T cell infusion, a product-specific cause would seem unlikely. When did the infections occur? Were the bacteria identified? Also, what was the nature of the infections – pneumonia, bacteremia, catheter-associated, etc.?”

Thank you for this very important and relevant comment. We would like to highlight that the US Immunotherapy Consortium reported infections in close to half of their patients treated with cilta-cel (47%) as well [13]. In their combined cohort from 16 sites, the majority of the infections were bacterial and viral.

It is true that we observed a somewhat higher rate of infections in our patients treated with cilta-cel which may be explained by differences in practices such as the routine use of IVIG and prophylactic antibiotics or differences in diagnosing and reporting infections at other institutions. At our institution, post-cilta-cel infections were primarily seen within 1-3 months after CAR T. Bacterial infections, which constitute the majority of infections in our study, were typically caused by gram-negative bacteria or *Clostridium difficile*.

In terms of the nature of the bacterial infections our cilta-cel patients had one or more of the following types: bacteremia (6/14; 43%), colitis (5/14; 36%), urinary tract infection (5/14; 36%), pneumonia (2/14; 14%), deep tissue infection (1/14; 7%). We have updated the Results section of our revised manuscript to reflect the new findings outlined above.

Major comment #3: “The authors note more double-positive CD4/CD8 T cells post-cilta cel (Figure 1). However, these cells constitute a very minor fraction of total lymphocytes and are of unclear clinical relevance. Of note, studies in other hematologic malignancies have identified CD4/CD8 double-negative CAR T cells (Anderson ND et al Nat Med 2023) associated with CAR T-cell persistence. Were double-negative populations observed in this dataset? Also, the relevance of CD127 expression changes (Supplemental Figure 8) is unclear. Due to the limited number of colors of the flow panel used, this may represent a decrease in CAR T cells with a Treg-like phenotype given the global reduction in CD127 expression.”

These are all very important and interesting points. We agree that studies in other hematologic malignancies have identified CD4/CD8 double-negative CAR T cells associated with enhanced CAR T cell persistence. In these studies, the double-negative CAR T cells were clearly identifiable by flow cytometry, they appeared months to years after the initial treatment, and they showed an exhausted-like memory state and a distinct transcriptional signature [14]. We did not observe a reproducible population of CD4/CD8 double-negative CAR T cells in our patients receiving BCMA CAR T, at least not during the comparably limited follow-up period.

However, as the reviewer also pointed out, we did indeed observe a very convincing and consistent increase in a specific and easily identifiable subpopulation of CAR T cells, CD4⁺ and CD8⁺ double-positive T cells, following treatment with cilta-cel. As the reviewer correctly stated, the clinical relevance of these cells has not been defined in great detail. However, there are some very interesting published findings on this T cell subpopulation which we would like to discuss. Some studies have indicated unique tumor-targeting properties of these types of T cells [15]. However, the same T cell subtype also plays a role in promoting autoimmune diseases [16] and Human T cell leukemia virus type 1 (HTLV-1)-associated CNS disease [17]. Importantly, a recent study identified a CD4⁺/CD8⁺ double-positive T cell (DPT) population, not present in starting grafts, in adult allo-HCT recipients. In these patients, the presence of DPT was predictive of \geq grade 2

GVHD. Furthermore, they showed that isolated DPTs were sufficient to mediate xeno-GVHD pathology when retransplanted into naïve mice but provided no survival benefit when mice were challenged with a human B-ALL cell line [18]. Overall, we consider it possible that in our patients this CAR T subpopulation could play a role in mediating anti-myeloma responses but also in promoting off-tumor toxicities like neurotoxicity. We have amended the Discussion part of the revised version of our manuscript to include a summary of all the additional information provided above.

Finally, we agree with the reviewer that the increase of the CD127 expression on the cilta-cel CAR T cells may alternatively represent a reduction in CAR T with a CD127^{low} T regulatory phenotype, we added a reference to the manuscript supporting this view, and we revised the Discussion section of the manuscript accordingly. However, we also note substantial published data supporting the alternative view - that this phenomenon may confer a survival advantage upon the cilta-cel CAR T cells.

Major comment #4: “The scRNAseq data in Figure 4 is from a single cilta cel patient. How/why was this patient chosen? This is not trivial, as factors such as prior therapies, depth of response, and age, can impact the findings. Of note, the scRNAseq data from Figure 4 demonstrate a significant increase in the CD8 EM population (red bar) when comparing panel C to panels D or E, but the change in the CD8 EM population was not statistically significant in Figure S4. This discrepancy makes interpretation of the data in Figure 4 problematic, as it is representative of a single patient that does not align with the trend of the cilta cel cohort. Also, a comparison with a matched ide cel patient(s) would be interesting/insightful.”

Thank you for raising this important point. Patient 663 was simply chosen because they represented one of the first cilta-cel patients enrolled into our GCCC2043 immunomonitoring study. Accordingly, this was one of the very first patients in whom we noticed this marked expansion of CAR T cells over the first few weeks that, in retrospect, turned out to take place in the majority of cilta-cel patients. The patient did not have any unusual clinical characteristics and more or less represents a typical cilta-cel patient as described in Table 1. We have clarified this in the revised version of our manuscript.

Regarding an apparent lack of correlation between RNAseq and flow data describing a shift in CAR T memory subtypes, we would like to clarify that there is always a potential discrepancy between the definition of certain memory subpopulations using their gene expression profile vs. their expression of certain pre-defined surface markers. We would also like to clarify that while Figure 4 describes changes in CAR T cell memory subtypes over the first 4 weeks after CAR T cell infusion, Supplemental Figure 4 describes our comparative analysis of CAR T cell memory subtypes at peak vs. a “late” timepoint at 3 months post treatment. Figure 3, on the other hand, does describe CAR T memory subpopulations at the individual peak level and it shows that cilta-cel patients evidence a higher level of CD8⁺ effector-memory-type CAR T cells than ide-cel patients. Finally, we would like to highlight that both figures describe an increase in certain effector-type CD8⁺ T cells, the difference being that Figure 4 shows a subgroup of CD8⁺ CAR-T cells defined as

“effector memory” according to their RNA signature, whereas Supplemental Figure 4 shows “terminal effector memory” CD8⁺ CAR T cells as defined by flow cytometry.

We agree that it would have been preferable to include a group of patients who received ide-cel vs. cilta-cel in our RNA sequencing analyses. Unfortunately, limited resources did not allow this and we had to focus on the group of patients with the more significant changes in CAR T cell numbers and composition over time, i.e. the patients who had received cilta-cel. That being said, we completely agree with the reviewer that our numbers in the RNAseq section are relatively low and, therefore, we decided to tone down our conclusions derived from these analyses in the revised version of the manuscript.

Major comment #5: “The authors state in the abstract that “comparably longer persistence of cilta cell CAR T was associated with increased resistance to exhaustion...” and subsequently in the results state “Remarkably, we found the highest levels of exhaustion markers in cilta cel and ide cel-treated patients across T cell subsets immediately after infusion. This initial upregulation was followed by a decrease in exhaustion markers over the following weeks.” This observation is not indicative of exhaustion, as transiently increased expression of negative regulatory markers (PD-1, LAGE-3, etc) is expected in the setting of T-cell activation in the early window after CAR T-cell infusion. Also, this observation does not signify resistance to exhaustion.”

We thank the reviewer for these salient points; we agree with the reviewer and in the revised version of the manuscript we now explain that, as the reviewer correctly stated, the transiently increased expression of negative regulatory markers is not necessarily indicative of exhaustion. We also revised our manuscript to make clear that the reduced impact of an “early” upregulation of negative regulatory factors on cilta-cel vs. ide-cel expansion, as the reviewer suggested, does not signify resistance to exhaustion per se.

Minor Comments

Minor comment #1: “The specific time points for sample collections should be included in the methods and/or results section.”

The reviewer raises a good point; in response to their critique we have added information on the specific timepoints at which the different sample types were collected to the revised version of the Methods section. We have also added some more general information on sample collection and processing.

Minor comment #2: “Additional details regarding methods of flow-cytometric-based analysis of CAR T-cell populations are needed (can be described in the supplement). While the authors provide information about anti-CAR antibodies in supplemental Table 1, the method of CAR T-cell identification is not shown beyond what was shown in Figure 3A. Were FMOs used or was the gate in Figure 3A drawn arbitrarily based on the fluorescence intensity histogram? The use of FMO on a population gated for this type of analysis is essential to accurately discriminate positive and negative cell populations.”

We agree with the reviewer and we are providing additional information on the flow-cytometric analysis of CAR T cell populations in the Methods section of the revised manuscript. For example, we are explaining that we have always used FMOs to reliably define positive versus negative cell populations. Importantly, we did not only use FMOs to define CAR-positive T cell populations but we also used them to assess expression of certain surface markers on the CAR T cells themselves such as CD27, CD127, TIM3, PD-1 and LAG-3. We had already included examples of this strategy in the original version of the manuscript (e.g., Supplemental Figure 8A) but, in response to the reviewer's comment, we have generated additional examples for the revised version of the manuscript (e.g., revised Supplemental Figure 1A).

Minor comment #3: “Comparisons between duration of response or total cohort progression-free survival should be included in Table 1, and not just overall survival in supplemental Figure 3. This is especially important to inform the analysis of the differential rates of toxicities between the two therapies that the authors report.”

We agree with the reviewer and have included data on PFS into the revised version of Table 1. Unfortunately, our original manuscript did not make sufficiently clear that Supplemental Figure 3D already showed PFS and not overall survival. In the revised version of the manuscript, we have clarified this. We do not think that an analysis of overall survival would be justified given the relatively small sample size and limited follow-up.

Minor comment #4: “Consider an alternative color to the green used in Figures 2D, S3B, S3E, S5A, and S5C, as when contrasted with adjacent similar greens and reds (see Figure S5), may prove difficult to perceive by readers with deuteranopia.”

We thank Reviewer 2 for providing us with the opportunity to improve the presentation of our findings. As a result, we have thoroughly revised figures 2D, S3B, S3E, S5A, and S5C; in the revised versions we use contrasting colors to more accessibly illustrate our results. Figure legends are revised accordingly.

Minor comment #5: “Supplemental Figure 1: a) Labeled as “Overexpression” of CD27.... “Higher” or “increased” would be more accurate/appropriate. b) The inclusion of a representative dot plot with gating is recommended.”

We agree with this reviewer; in the title of the revised version of Supplemental Figure 1 we describe the expression of CD27 as “increased” instead of “overexpressed”. As briefly mentioned above, in the representative cilta-cel patient 305, we added a figure to illustrate our gating strategy and assessment of CD27 surface expression on CAR-T cells using an FMO control (revised Supplemental Figure 1A).

Minor comment #6: “References: #5 & 6 need to be reformatted.”

The revised manuscript includes appropriately reformatted references.

REVIEWERS 3 AND 4

We would like to thank both reviewers for highlighting the strengths of our report without requiring any additional edits.

References

1. Oriol, A., et al., *The role of idecabtagene vicleucel in patients with heavily pretreated refractory multiple myeloma*. Ther Adv Hematol, 2021. **12**: p. 20406207211019622.
2. Montes de Oca, R., et al., *Biomarker Correlates of Response to Ciltacabtagene Autoleucel in Patients with Relapsed or Refractory Multiple Myeloma from CARTITUDE-1, a Phase 1b/2 Open-Label Study, at the ~3 Year Follow-up*. Blood, 2023. **142**(Supplement 1): p. 2099-2099.
3. Zhou, J., et al., *The persistence and antitumor efficacy of CAR-T cells are modulated by tonic signaling within the CDR*. Int Immunopharmacol, 2024. **126**: p. 111239.
4. Pipkin, M.E., et al., *Interleukin-2 and inflammation induce distinct transcriptional programs that promote the differentiation of effector cytolytic T cells*. Immunity, 2010. **32**(1): p. 79-90.
5. Mehta, A.K., D.T. Gracias, and M. Croft, *TNF activity and T cells*. Cytokine, 2018. **101**: p. 14-18.
6. Andrews, J.S., A.E. Berger, and C.F. Ware, *Characterization of the receptor for tumor necrosis factor (TNF) and lymphotoxin (LT) on human T lymphocytes. TNF and LT differ in their receptor binding properties and the induction of MHC class I proteins on a human CD4+ T cell hybridoma*. J Immunol, 1990. **144**(7): p. 2582-91.
7. Aspalter, R.M., M.M. Eibl, and H.M. Wolf, *Regulation of TCR-mediated T cell activation by TNF-RII*. J Leukoc Biol, 2003. **74**(4): p. 572-82.
8. Yokota, S., T.D. Geppert, and P.E. Lipsky, *Enhancement of antigen- and mitogen-induced human T lymphocyte proliferation by tumor necrosis factor-alpha*. J Immunol, 1988. **140**(2): p. 531-6.
9. Ranges, G.E., et al., *Tumor necrosis factor alpha/cachectin is a growth factor for thymocytes. Synergistic interactions with other cytokines*. J Exp Med, 1988. **167**(4): p. 1472-8.
10. *FDA approves idecabtagene vicleucel for multiple myeloma*. 2021 02/20/2025]; Available from: <https://www.fda.gov/drugs/resources-information-approved-drugs/fda-approves-idecabtagene-vicleucel-multiple-myeloma>.
11. *FDA approves ciltacabtagene autoleucel for relapsed or refractory multiple myeloma*. 2022 02/20/2025]; Available from: <https://www.fda.gov/drugs/resources-information-approved-drugs/fda-approves-ciltacabtagene-autoleucel-relapsed-or-refractory-multiple-myeloma>.
12. Wu, J., et al., *Clinical Pharmacological Characterization of Cilta-cel, a CAR-T Therapy Directed Against BCMA in Adult Patients with Multiple Myeloma in a Multicohort CARTITUDE-2 Study*, in *2024 American College of Clinical Pharmacology (ACCP) Annual Meeting*. 2024: Bethesda, Maryland.
13. Sidana, S., et al., *Safety and efficacy of standard-of-care ciltacabtagene autoleucel for relapsed/refractory multiple myeloma*. Blood, 2025. **145**(1): p. 85-97.

14. Anderson, N.D., et al., *Transcriptional signatures associated with persisting CD19 CAR-T cells in children with leukemia*. Nat Med, 2023. **29**(7): p. 1700-1709.
15. Alam, M.R., et al., *CD4(+)CD8(+) double-positive T cells in immune disorders and cancer: Prospects and hurdles in immunotherapy*. Autoimmun Rev, 2025. **24**(3): p. 103757.
16. Xu, H.M., et al., *Epigenetic DNA methylation of Zbtb7b regulates the population of double-positive CD4(+)CD8(+) T cells in ulcerative colitis*. J Transl Med, 2022. **20**(1): p. 289.
17. Maher, A.K., et al., *HTLV-1 induces an inflammatory CD4+CD8+ T cell population in HTLV-1-associated myelopathy*. JCI Insight, 2024. **9**(1).
18. Hess, N.J., et al., *Inflammatory CD4/CD8 double-positive human T cells arise from reactive CD8 T cells and are sufficient to mediate GVHD pathology*. Sci Adv, 2023. **9**(12): p. eadf0567.

As requested, please find below our point-by-point response to all comments:

REVIEWER 2

Comment #1: “In the Abstract, the authors write: ‘The pronounced expansion of cilta-cel CAR T was associated with higher CAR and CD27 expression and reduced impact of upregulation of negative regulatory factor TIM3 on the infused cells.’ => The data show a ‘decrease’ in TIM-3 expression, but conclusions about the functional significance or ‘impact’ of this observation are speculative. This should be revised to ensure the findings are not overstated.”

Thank you very much for this comment. In the revised version of the manuscript, we have modified the abstract which now states that there was a lack of an association between TIM3 expression and CAR T cell proliferation in the cilta-cel group. We do not propose any potential causal relationships anymore.

Comment #2: “In the Discussion (page 16, 2nd paragraph), the authors write: ‘Importantly, we found only in the ide-cel group that the ‘early’ peak in expression of the exhaustion marker TIM3, or of all three markers combined, predicted reduced in-vivo CAR T proliferation a few days later. There was no such association in the cilta-cel group indicating that the freshly produced cilta-cel cells may be more resistant to the immediate impact of negative regulatory factor TIM3.’ => Similar to comment #1, this section should be revised as TIM-3 expression alone is not a definitive marker of exhaustion. Additionally, it is not accurate to conclude that cilta-cel cells are ‘resistant’ to the ‘immediate impact of the negative regulatory factor TIM-3.’ This statement overinterprets the data and should be reworded for accuracy.”

Thank you for bringing up this point. In response to the reviewer’s comment, we have removed any reference to TIM3 as an exhaustion marker from the section mentioned above. We have also removed our conclusion that our findings may indicate a reduced functional impact on the cilta-cel CAR T cells.

Comment #3: “In the Discussion (page 17, 1st paragraph, last sentence), the authors write: ‘...although it may also represent a reduction in CAR T with a CD127^{low} T regulatory phenotype, most likely confers a survival advantage on the cilta-cel cells.’ => ‘...most likely confers a survival advantage...’ needs to be revised to ‘may confer a survival advantage.’”

We agree with the reviewer, and we have revised the manuscript as suggested.

Comment #4: “In the Figure Legend 5 (page 27), the title of ‘Cilta-cel CAR-T show short-term resistance to exhaustion with an upregulation of exhaustion markers on long-term persisting cells’ needs to be revised to remove ‘resistance to exhaustion’ and change ‘exhaustion markers’ to ‘negative regulatory markers’”.

Thank you for this comment. We have revised the title of Figure Legend 5 according to the reviewer's suggestions.

Comment #5: *“In the Figure Legend 5 (page 27): The multiple references to ‘exhaustion markers’ should be changed to ‘negative regulatory markers.’”*

In response to the reviewer's comment, we have revised Figure Legend 5 as suggested.

Comment #6: *“Results section page 12 on ‘Long-term persistence...with upregulation of exhaustion markers and IL-7Ra.’ => The phrase ‘exhaustion markers’ is inaccurate, as transiently increased expression of negative regulatory markers (e.g., PD-1, LAG-3) is an expected response to T-cell activation in the early period following CAR T-cell infusion. Recommend rewording this section, as the authors have not demonstrated T-cell exhaustion and may be overstating their findings.”*

We agree with the reviewer and have revised this section of the manuscript accordingly.

Comment #7: *“Response to major comment #4: Presenting this data in a main figure is questionable and potentially misleading, given that it is based on a single patient. As previously noted, this raises substantial concerns about its limitations and lack of generalizability.”*

As recommended by the reviewer, we removed the previous “Figure 4” as a main figure and added it as the new “Supplemental Figure 3”. We present the previous “Supplemental Figure 3”, containing primarily clinical data, as a main figure (new “Figure 3”) instead. The sequence of figures was adapted accordingly throughout the manuscript.